# Massive normalization of olfactory bulb output in mice with a 'monoclonal nose'

Benjamin Roland[1†], Rebecca Jordan[2,3†], Dara L Sosulski[4†], Assunta Diodato[1], Izumi Fukunaga[2,5], Ian Wickersham[6], Kevin M Franks[7], Andreas T Schaefer[2,3,5,8], Alexander Fleischmann[1*]

[1]Center for Interdisciplinary Research in Biology, Collège de France, INSERM U1050, CNRS UMR 7241, Paris, France; [2]The Francis Crick Institute, London, United Kingdom; [3]Department of Neuroscience, Physiology and Pharmacology, University College London, London, United Kingdom; [4]Wolfson Institute for Biomedical Research, University College London, London, United Kingdom; [5]Behavioural Neurophysiology, Max-Planck-Institute for Medical Research, Heidelberg, Germany; [6]MIT Genetic Neuroengineering Group, McGovern Institute for Brain Research, Massachusetts Institute of Technology, Cambridge, United States; [7]Department of Neurobiology, Duke University, Durham, United States; [8]Department of Anatomy and Cell Biology, Faculty of Medicine, University of Heidelberg, Heidelberg, Germany

*For correspondence: alexander.
fleischmann@college-de-france.fr

†These authors contributed
equally to this work

Reviewing editor: Upinder S
Bhalla, National Centre for
Biological Sciences, India

**Abstract** Perturbations in neural circuits can provide mechanistic understanding of the neural correlates of behavior. In M71 transgenic mice with a "monoclonal nose", glomerular input patterns in the olfactory bulb are massively perturbed and olfactory behaviors are altered. To gain insights into how olfactory circuits can process such degraded inputs we characterized odor-evoked responses of olfactory bulb mitral cells and interneurons. Surprisingly, calcium imaging experiments reveal that mitral cell responses in M71 transgenic mice are largely normal, highlighting a remarkable capacity of olfactory circuits to normalize sensory input. In vivo whole cell recordings suggest that feedforward inhibition from olfactory bulb periglomerular cells can mediate this signal normalization. Together, our results identify inhibitory circuits in the olfactory bulb as a mechanistic basis for many of the behavioral phenotypes of mice with a "monoclonal nose" and highlight how substantially degraded odor input can be transformed to yield meaningful olfactory bulb output.

## Introduction

Odorants in the environment are detected by a large repertoire of odorant receptor, expressed on the dendrites of sensory neurons in the olfactory epithelium (*Buck and Axel, 1991*; *Zhang and Firestein, 2002*). In mice, each olfactory sensory neuron expresses only one of ~1300 odorant receptor genes, and each of these receptors interacts with multiple odorants (*Chess et al., 1994*; *Malnic et al., 1999*). Neurons expressing a given receptor are distributed randomly across large zones of the olfactory epithelium, but project to two spatially invariant glomeruli in the olfactory bulb, the first processing center of olfactory information in the mammalian brain (*Ressler et al., 1994*; *Vassar et al., 1994*). Thus, the distributed pattern of neural activity that is evoked by the binding of an odorant to a given receptor in the olfactory epithelium is transformed into a topographically organized, invariant map of glomerular activity at the level of the olfactory bulb (*Bozza et al., 2004*; *Meister and Bonhoeffer, 2001*; *Rubin and Katz, 1999*; *Uchida et al., 2000*; *Wachowiak and Cohen, 2001*).

**eLife digest** The lining of the nose contains cells called olfactory sensory neurons that allow different smells to be detected. Odor molecules bind to receptor proteins that are embedded in the surface of the olfactory sensory neuron. Different receptors respond to different odors, and the nose contains hundreds of different receptors that work together to distinguish thousands of scents. When an odor molecule binds to a receptor, it triggers a pattern of electrical activity in the neuron. These patterns are the building blocks that allow smells to be recognized and if necessary, acted upon – by not eating food that smells rancid, for example.

In 2008, researchers genetically engineered mice so that nearly all of their olfactory sensory neurons produced the same type of olfactory receptor. Unexpectedly, these mice could still detect and discriminate between many different smells. Now, Roland, Jordan, Sosulski et al. – including several of the researchers involved in the 2008 study – have tracked the brain activity of these mice as they were exposed to various smells to find out how they can recognize such a wide range of odors with such a limited repertoire of receptors.

The results of the experiments revealed that neural circuits in the brains of these modified mice still produce largely normal patterns of activity in response to an odor. This 'normalization' of activity relies on a fine balance between 'excitatory' processes that increase the activity of neurons and 'inhibitory' processes that reduce this activity.

Overall, the findings of Roland, Jordan, Sosulski et al. provide a link between how a scent is detected and how this information is processed in the brain. In future experiments, it will be important to determine how this processing of odor information is influenced by learning and experience to generate the long-lasting odor memories that guide behavior.

The principal neurons of the olfactory bulb, mitral and tufted cells, extend their apical dendrite into a single glomerulus, and thus only receive direct input from sensory neurons expressing a single odorant receptor. Electrophysiological and imaging experiments have revealed that, consistent with this anatomical organization, mitral cells tend to be narrowly tuned and only respond to a small number of odorants (*Davison and Katz, 2007*; *Tan et al., 2010*). The spatiotemporal patterns of mitral cell firing are strongly shaped by the activity of local inhibitory neurons, including periglomerular cells, EPL interneurons, and granule cells (*Banerjee et al., 2015*; *Fukunaga et al., 2014*; *Kato et al., 2013*; *Luo and Katz, 2001*; *Miyamichi et al., 2013*; *Yokoi et al., 1995*). Ultimately, mitral and tufted cells relay this odor information to several higher brain regions, including the piriform cortex, amygdala, and entorhinal cortex, via a dense elaboration of axonal projections (*Ghosh et al., 2011*; *Igarashi et al., 2012*; *Luskin and Price, 1982*; *Miyamichi et al., 2011*; *Nagayama et al., 2010*; *Sosulski et al., 2011*). How the patterns of activity evoked by odor stimulation in the cells of the olfactory bulb ultimately relate to odor perception, discrimination, and behavior, however, remains largely undefined.

A major challenge for the olfactory system is that it must function across a wide range of stimulus intensities. For example, salient odor cues must reliably be detectable against strong and highly dynamic background odors. To explore potential neural mechanisms that can mediate such signal amplification and noise reduction we used previously generated M71 transgenic mice with a 'monoclonal nose' (*Fleischmann et al., 2008*). In these mice, more than 95% of all olfactory sensory neurons express a single odorant receptor, M71. As a consequence of this genetic manipulation, the frequency of sensory neurons expressing endogenous odorant receptor genes is reduced 20-fold, and the canonical glomerular odor map observed in wild-type mice disappears: most odorants now fail to elicit detectable levels of glomerular activity, while the majority of glomeruli respond to acetophenone, a known M71 ligand (*Figure 1—figure supplement 1*).

Surprisingly, despite this striking alteration of odor-evoked neural activity, M71 transgenic mice are able to smell a variety of odors. They can detect and discriminate several odorants in a go/no go operant conditioning task, although their performance in this task decreases compared to controls when M71 transgenic mice are required to discriminate mixtures of structurally and perceptually similar odorants. Moreover, M71 transgenic mice fail to discriminate acetophenone, a known strong

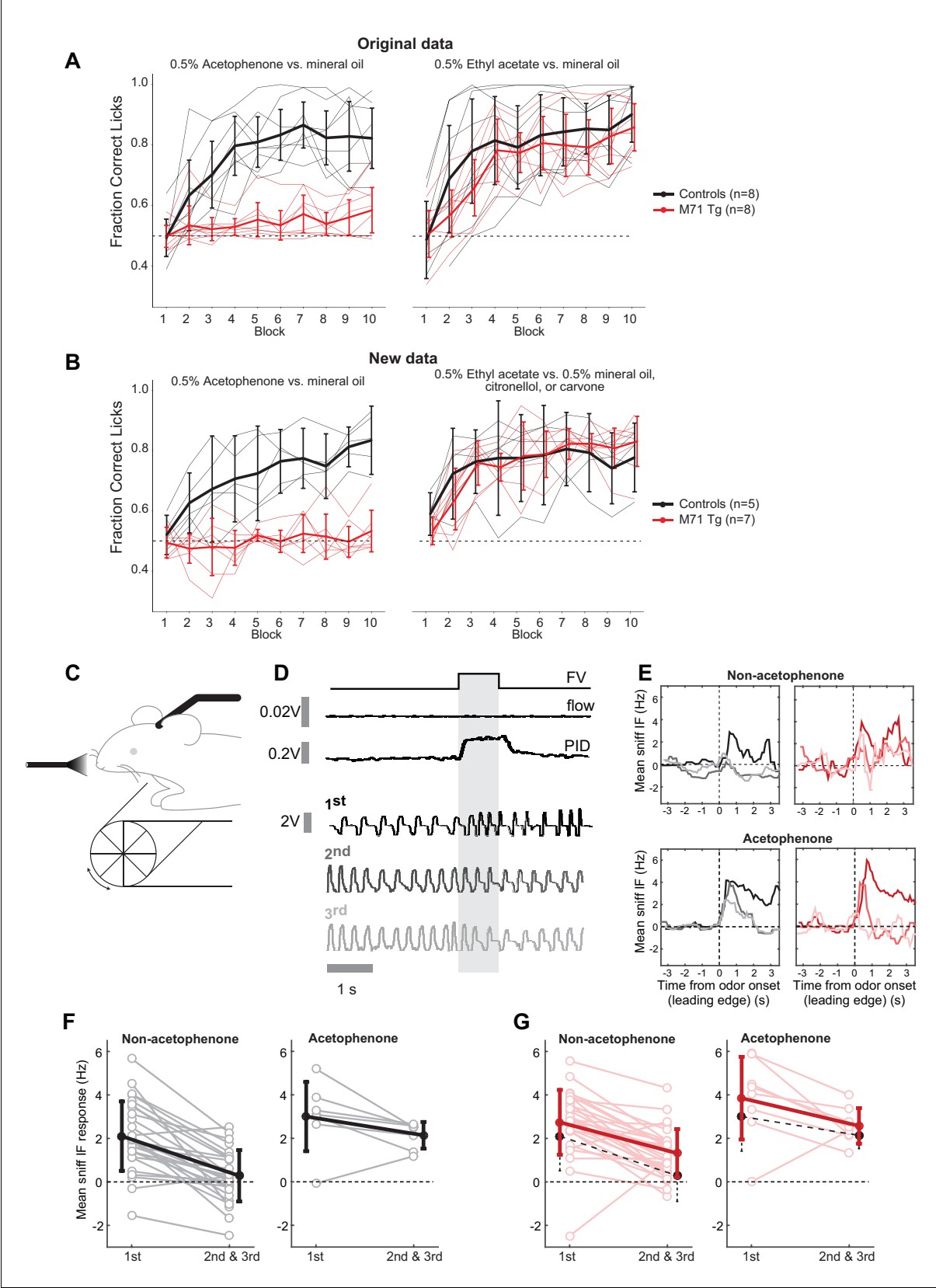

**Figure 1.** The ability of M71 transgenic mice to detect acetophenone is task-dependent. (A, B) In a go/no go operant conditioning task, M71 transgenic mice fail to discriminate acetophenone from mineral oil (left panels). In contrast, M71 transgenic mice readily discriminate other pairs of

*Figure 1 continued on next page*

*Figure 1 continued*

odorants (ethyl acetate vs. mineral oil, citronellol, or carvone, right panels). (**A**) Original results reported in *Fleischmann et al. (2008)*. (**B**) Repeat experiment with an additional cohort of mice. Thin lines: learning curves for individual mice. Thick lines: averaged learning curves. Error bars: 95% CI of the mean. (**C**) Sniff adaptation: schematic of the experimental configuration. (**D**) Example sniff traces during first 3 (1st, 2nd, and 3rd) presentations of hexanone (shaded area) from a control mouse. Lighter colored traces signify later presentations. 'FV' trace shows opening of final valve directing odorized air to the mouse, 'flow' trace shows the output from the olfactometer, and 'PID' trace shows signal evoked by odorized air from a photo-ionization detector. (**E**) Example moving averages of instantaneous sniff frequency during first 3 presentations of hexanone (window = 500 ms, plotted against leading edge). Black traces: controls, red traces: M71 transgenic mice. (**F, G**) Mean instantaneous sniff frequency responses to first vs. the average of the 2nd and 3rd presentation of an odor for control (black, **F**) and M71 transgenic (red, **G**) mice. Pooled non-acetophenone odorants: hexanone, ethyl acetate, heptanal, and an odor mixture. Lighter colors: individual trials, thick lines: averages. Error bars: SD. Black dotted lines on the M71 plots show the means for the corresponding data from controls.

The following figure supplements are available for figure 1:

**Figure supplement 1.** Schematic representation of the perturbation of the glomerular map of M71 transgenic mice with a 'monoclonal nose'.

**Figure supplement 2.** M71 transgenic mice fail to detect acetophenone in a go/no go operant conditioning task.

M71 ligand, from air in this go/no go discrimination assay, despite the fact that acetophenone activates the vast majority of sensory neurons and glomeruli in these mice.

This apparent discrepancy between molecular alteration and receptor neuron physiology on the one side and behavioral phenotype on the other now allows us to investigate the neural mechanisms at play: What does allow M71 transgenic mice to detect and discriminate most odorants despite the 20-fold decrease in the expression of endogenous odorant receptors? Conversely, what underlies the inability of these mice to detect the pervasive glomerular activity evoked by acetophenone? To explore the link between odor-evoked sensory neuron activity and behavior we analyzed the activity of olfactory bulb mitral cells, the main output neurons of the olfactory bulb. Two-photon calcium imaging and whole cell patch-clamp recordings of mitral cells revealed that mitral cell odor responses in M71 transgenic mice greatly resembled the responses observed in wild-type mice. Indeed, the fraction of responsive mitral cells and odor-evoked changes in firing rates were indistinguishable from controls. Calcium imaging and whole cell recordings further indicated that much of this normalization of odor-evoked activity is achieved through inhibition by periglomerular interneurons. Finally, we found that M71 transgenic mice exhibit spontaneous sniff adaptation in response to acetophenone exposure, suggesting that while they consistently fail to discriminate acetophenone from air in a go/no-go operant conditioning tasks they are indeed able to detect the presence of acetophenone. Together, our data reveal that odor-evoked patterns of glomerular activity can be substantially transformed by olfactory bulb neural circuits, to extract meaningful odor information from massively degraded sensory input and point towards a key role of glomerular inhibition.

## Results

### The ability of M71 transgenic mice to detect acteophenone is task-dependent

Previous behavioral experiments using a go/no go operant conditioning task indicated that M71 transgenic mice failed to discriminate acetophenone from its diluent, mineral oil, but could detect and discriminate other odorants (*Fleischmann et al., 2008*) (*Figure 1—figure supplement 1*). To better understand the link between odor-evoked neural activity and behavior we first replicated these behavioral observations with an independent cohort of mice. Consistent with our initial observations, we found that M71 transgenic mice consistently failed to detect acteophenone in this task (acteophenone versus mineral oil, repeated measure ANOVA, (block x genotype) $F_{(9,90)} = 5.43$, p<0.001), yet readily discriminated other pairs of odorants (ethyl acetate versus mineral oil, citronellol, or carvone, (block x genotype) $F_{(9,90)} = 1.49$, p=0.17, *Figure 1A and B*). Individual experiments consisted of 10 blocks of 20 odor presentations, and all 15 M71 transgenic mice failed to reach a 'correct lick ratio' surpassing 75%. In contrast, the same 15 M71 transgenic mice all successfully discriminated ethyl acetate from mineral oil, citronellol, or carvone (*Figure 1A and B*, *Figure 1—figure*

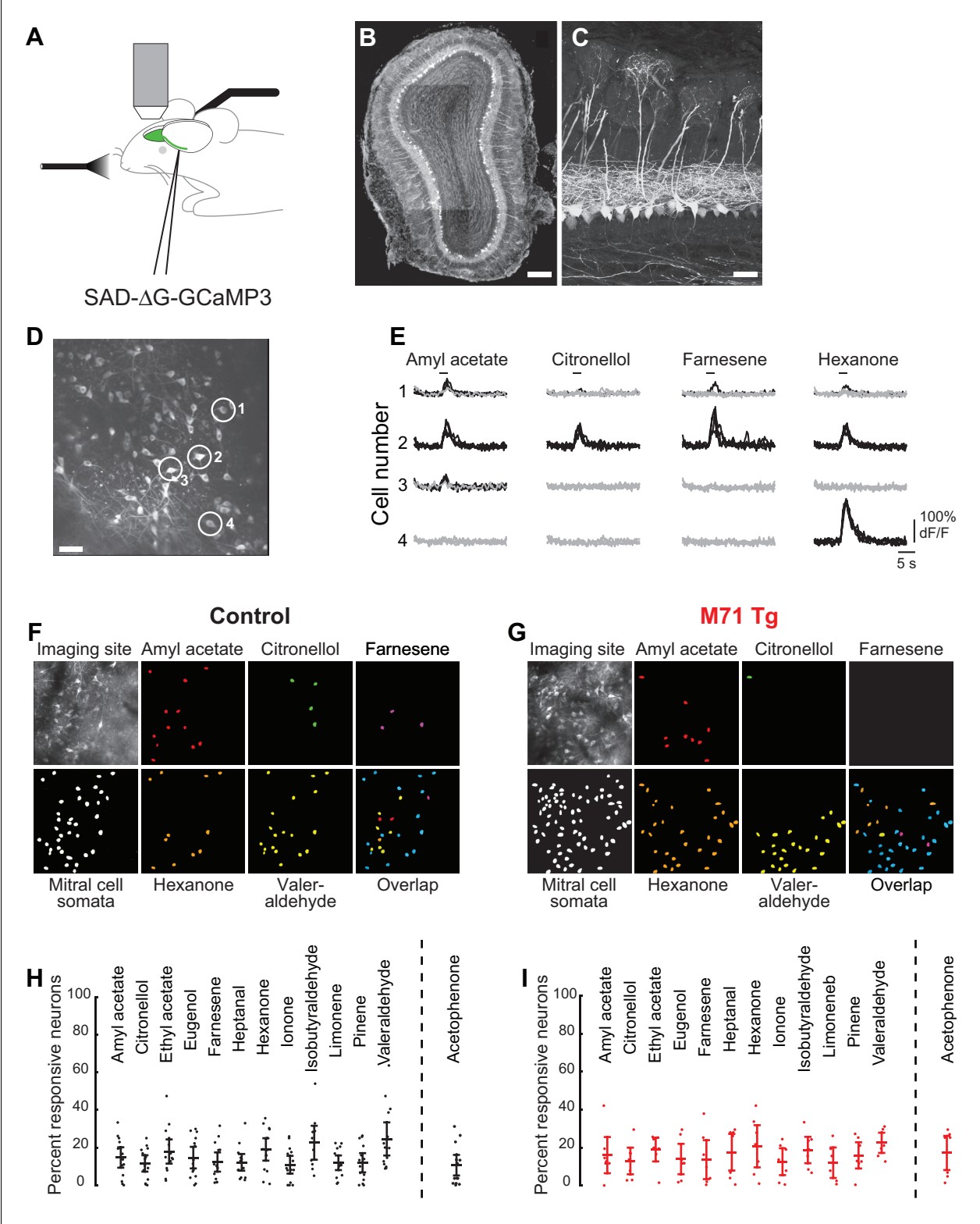

**Figure 2.** Normalization of odor-evoked mitral cell activity in M71 transgenic mice. (**A–I**) Two-photon in vivo imaging of mitral cell odor responses in anesthetized mice. (**A**) Schematic of rabies-GCaMP3 injection into the lateral olfactory tract (LOT) and two-photon imaging of olfactory bulb mitral cells. (**B**) Mitral cells expressing GCaMP3 in a coronal slice of the olfactory bulb after injection of rabies-GCaMP3. Note the restriction of labeled cell bodies to the mitral cell layer. Scale bar = 100 μM. (**C**) Higher magnification of mitral cells expressing GCaMP3 throughout the neuron, including the apical and lateral dendrites. Scale bar = 20 μM. (**D**) Two-photon micrograph showing GCaMP3 expression in mitral cell of a single imaging site. Scale bar = 30 μM.
*Figure 2 continued on next page*

*Figure 2 continued*

(E) Example traces of the responses of 4 mitral cells (circled in (D)) to 4 different odorants. Traces represent responses to 4 individual odorant exposures, non-responsive trials are shown in grey, responsive trials in black. Horizontal bar indicates odorant application. (F, G) Representative maps of odor-evoked mitral cell activity elicited by a panel of 5 different odorants at a single imaging site in a control (F) and M71 transgenic (G) mouse. Cells responding to at least 2 out of 4 trails are color-coded. Overlap: cells responsive to more than one odorant are shown in blue. (H, I) Mean fraction of cells (horizontal line) responding to a given odorant at 0.01% vol./vol. dilution, in control (H) and M71 transgenic (I) mice. Dots represent the fraction of responding cells for a given imaging site. Controls: 14 imaging sites in 7 mice, n (median number of cells per site) = 35; M71 transgenic: 7 imaging sites in 4 mice, n = 28. Error bars = 95% CI of the mean.

*supplement 2*). Thus, in a go/no-go operant conditioning task, M71 transgenic mice are consistently unable to discriminate acetophenone from air.

We next asked if the failure to detect acetophenone was specific for this go/no go operant conditioning task, or whether it could similarly be observed in a different behavioral test. To address this question we measured exploratory sniffing behavior in response to novel odors, a simple, spontaneous test for odor perception, which does not require training (*Welker, 1964*; *Wesson et al., 2008*). As previously described, wild-type mice exhibited increased sniff frequencies when exposed to a novel odorant (mean response = 2.1 Hz, SD = 1.6 Hz) that then decreased upon repeated presentation of the same odorant (mean response for 2nd and 3rd presentation = 0.3 Hz, SD = 1.2 Hz, p=3.3 $\times 10^{-7}$, paired t-test 1st presentation versus mean of 2nd and 3rd, n = 27 mouse-odor pairs, n = 7 mice, *Figure 1C–F*). Consistent with their ability to detect and discriminate most odorants, M71 transgenic mice exhibited an initial increase in sniff frequency to ethyl acetate, hexanone, heptanal, or a mixture of isoamyl acetate, 2-nonanone and cyclohexanol (referred to as 'non-acetophenone' odors in *Figure 1F*, mean response = 2.7 Hz, SD = 1.4 Hz). This response was indistinguishable from controls (p=0.12, t-test), and similarly displayed a significant decrease in sniff frequencies upon re-exposure (p=1.7 $\times 10^{-8}$, paired t-test, n = 29 mouse-odor pairs, n = 8 mice, *Figure 1G*). Surprisingly, similar results were obtained for acetophenone presentations: both control and M71 transgenic mice displayed initial high frequency responses (control: mean = 3.0 Hz, SD = 1.6 Hz; M71 transgenic: mean = 3.8 Hz, SD = 1.9 Hz, p=0.38, t-test), and reductions in response frequencies during the second and third acetophenone presentation (*Figure 1F and G*, right panels).

Together, these results indicate that in contrast to the aforementioned go/no go operant conditioning task, M71 transgenic mice can identify acetophenone in a spontaneous test for odor detection. Thus, in M71 transgenic mice the strong M71 ligand acetophenone results in major behavioral perturbations - while acetophenone is spontaneously detected, it cannot reliably be discriminated from background in an operant conditioning experiment.

## Rabies virus-mediated expression of GCaMP3 in olfactory bulb mitral cells

The ability to probe the cellular processes that underlie changes in olfactory-driven behaviors in this massively perturbed system can provide important general insights into how odor information is normally processed in the olfactory bulb. We therefore next asked how perturbed glomerular inputs in M71 transgenic mice are transformed into olfactory bulb outputs. We developed an in vivo imaging approach that permits the visualization of odor-evoked responses specifically in mitral cells, the main output neurons of the olfactory bulb. We used a replication-deficient recombinant rabies virus to drive the expression of the calcium-sensitive indicator of neural activity GCaMP (RVΔG-4GCaMP3) (*Tian et al., 2009*; *Wickersham et al., 2010*) in large populations of mitral cells. We made multiple injections of this rabies-GCaMP3 virus into the olfactory cortex underneath the lateral olfactory tract (*Figure 2A*). After injections, mice were allowed to recover for 5–7 days before two-photon imaging of neural activity was performed under ketamine/xylazine anesthesia. Because this modified rabies virus lacks the gene encoding its viral glycoprotein, it is unable to spread transsynaptically, thereby restricting the expression of GCaMP3 to neurons directly infected via their axonal terminations (*Wickersham et al., 2007*; *2010*). However, because the virus retains its ability to replicate in infected cells, we found that infected cells began exhibit clear signs of toxicity after 12 days (not shown). We therefore performed all imaging experiments within 5–7 days after virus injections.

Using this method, we were able to selectively express GCaMP3 in hundreds of mitral cells in the olfactory bulb (*Figure 2B–D*). GCaMP3-expressing mitral cells were uniformly distributed across the olfactory bulb. The cell bodies of GCaMP3-expressing neurons were exclusively located in the mitral cell layer, and we often observed multiple GCaMP3-expressing mitral cells projecting to the same glomerulus (*Figure 2C*). While we cannot exclude the possibility that GCaMP3 is also expressed in some tufted cells, these results demonstrate that rabies-GCaMP3 virus permits the highly efficient and selective labeling of mitral cells projecting to the piriform cortex.

## Similar odor-evoked responses in mitral cells of wild-type and M71 transgenic mice

Mitral cells infected with rabies-GCaMP3 displayed robust stimulus-locked responses to odorants, which could vary with respect to their response magnitudes (e.g. peak ΔF/F values), duration and trial-to-trial variability (*Figure 2E*). In wild-type mice, we found that odorants at low concentrations (0.01%, or 1/10,000 vol./vol. dilution) typically evoked sparse, spatially distributed patterns of activity in ~15% of mitral cells (mean = 14.5%, standard deviation (SD) = 11.2%; *Figure 2F and H*). These observations are consistent with recent results using adeno-associated virus (AAV)-mediated and transgenic GCaMP3 expression (*Blauvelt et al., 2013*; *Kato et al., 2012*; *Wachowiak et al., 2013*). We observed mitral cell responses to a variety of structurally and perceptually diverse odorants, regardless of whether the neurons were located in the posterior, medial, or anterior dorsal olfactory bulb (13 odorants at 0.01% vol./vol. dilution; *Figure 2F and H*, and data not shown). Furthermore, mitral cells responsive to a given odorant were typically distributed across the imaging site and did not exhibit the segregated patterns observed in odor-evoked glomerular activity.

We next performed these same imaging experiments using M71 transgenic mice. Remarkably, we found that the fraction and distribution of odor-responsive mitral cells in M71 transgenic mice and their wild-type littermate controls were strikingly similar (*Figure 2I*). Interestingly, our test set of odorants includes several odorants that have been reported to not activate the M71 receptor. Ethyl acetate or eugenol, for example, do not activate M71-expressing olfactory sensory neurons at all concentrations tested (*Bozza et al., 2002*), and do not elicit detectable glomerular activity in M71 transgenic mice (*Fleischmann et al., 2008*). However, we found that all test odorants including ethyl acetate and eugenol evoked mitral cell responses. Moreover, the fractions of odor-responsive neurons were indistinguishable from wild-type littermate controls (mixed-effect ANOVA (genotype x odorants), $F_{(13, 242)}$ = 0.58, p=0.87, *Figure 2H and I*). Thus, at least at the gross level of overall activation, different odorants including odorants that barely engage the large population of M71 expressing olfactory sensory neurons, result in the excitation of a population of mitral cells that is similar to wild-type mice.

Given that 95% of all olfactory sensory neurons in these mice express the M71 receptor, we next examined mitral cell responses to acetophenone, a known strong M71 receptor ligand (*Bozza et al., 2002*) that evoked pervasive glomerular activation of the dorsal surface of the olfactory bulb of M71 transgenic mice (*Fleischmann et al., 2008*). We observed two surprising findings: first, the fraction of mitral cells activated by acetophenone was virtually identical to littermate controls, and second, the fraction of mitral cells responding to acetophenone was highly similar to the fractions of mitral cells responding to other odorants (*Figure 2H,I* right). Given the massively altered glomerular input and essentially normal mitral cell output, these imaging data indicate that the OB circuitry profoundly normalizes activity, strengthening the weakened input from odorants that do not activate the M71 receptor, and suppressing the overt excitation due to the M71 ligand acetophenone, resulting in responses that – on the crude level of overall activation – were indistinguishable from wild-type mice.

A more detailed analysis of our imaging data, however, did reveal subtle differences between the response properties of M71 transgenic mitral cells and wild-type littermate controls. In controls, individual mitral cells generally displayed narrow stimulus tuning at low odor concentrations, in accord with previously published results from electrophysiological and optical recordings (*Figure 3A*) (*Davison and Katz, 2007*; *Kato et al., 2012*; *Tan et al., 2010*). Approximately half of the neurons (46.1%) did not respond to any of the 13 odorants in the stimulus set used to probe selectivity, while the majority of odor-responsive neurons (43.9% of all neurons) displayed significant increases in fluorescence to only 1–5 odor stimuli. A small subpopulation (10.0%) of mitral cells were more broadly tuned. Moreover, the majority of stimulus-evoked mitral cell response magnitudes were small, with

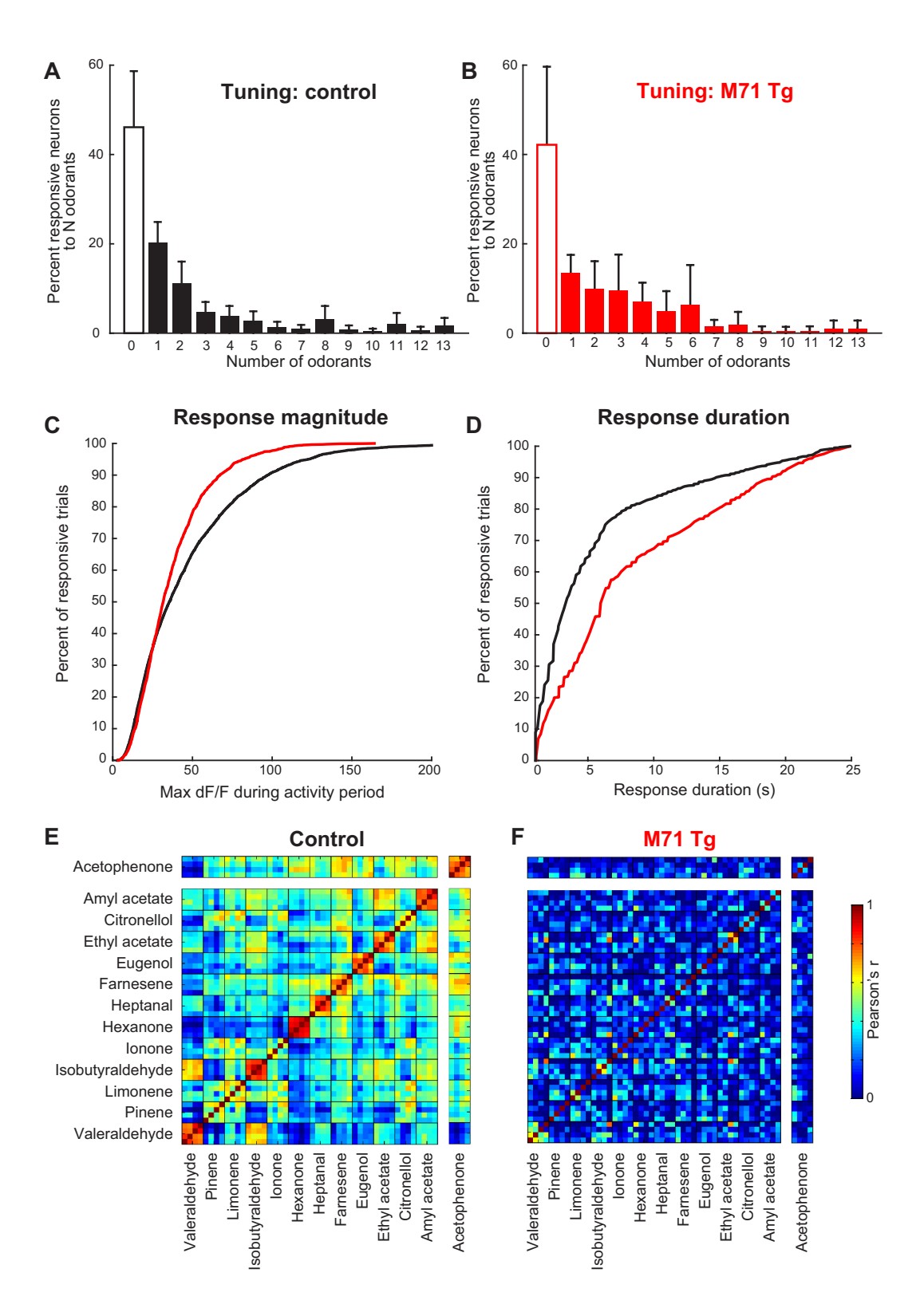

**Figure 3.** Normalization of odor-evoked neural activity in M71 transgenic mice results in changes in response magnitudes and duration, and trial-to-trial variability. (A–F) Two-photon in vivo imaging of mitral cell odor responses in anesthetized mice. (A, B) Odor tuning: the fraction of mitral cells

*Figure 3 continued on next page*

Figure 3 continued

responding to N odorants out of the 13 odorant test panel in control (**A**, black) and M71 transgenic (**B**, red) mice. Error bars = 95% CI. (**C**) Peak DF/F values for odor-evoked responses in control (black) and M71 transgenic (red) mice. The fraction of responses with high peak DF/F values is reduced in M71 transgenic mice. (**D**) Response durations are increased in M71 transgenic mice (red) compared to controls (black). (**E, F**) Trial-to-trial variability. Pearson's correlation coefficients for individual odor presentations (13 odorants, 4 trials per odorant). The similarities of response patters to 4 presentations of the same odorant is reduced in M71 transgenic mice.

The following figure supplement is available for figure 3:

**Figure supplement 1.** Response magnitudes and durations, and trial-to-trial variability of mitral cell odor responses in M71 transgenic mice.

peak $\Delta F/F$ values below 35%, although we could observe a small number of large stimulus-evoked responses with $\Delta F/F$ values of up to 200% (*Figure 3C*). Finally, the fluorescence levels of most odor-responsive neurons (>80%) returned to baseline within 8 s after response onset (*Figure 3D*). In M71 transgenic mice, mitral cells tended to be more broadly tuned (*Figure 3A and B*), but this difference did not reach statistical significance (Chi-squared test: $\chi 2$ = 17.7, p=0.17). For acetophenone and the 12 other odorants, the distribution of the response magnitudes of mitral cell was shifted towards smaller peak $\Delta F/F$ values, with a particularly large reduction in the number of strongly responding neurons (i.e. those reaching $\Delta F/F$ values of greater than 50%; *Figure 3C*, median $\Delta F/F$: control: 0.36, M71 transgenic: 0.32; mean $\Delta F/F$: control: 0.47, M71 transgenic: 0.38, Two-sample Kolmogorov-Smirnov test: $D_{4116,1639}$ = 0.12, p<0.01, and *Figure 3—figure supplement 1*). In contrast, the average response duration of M71 transgenic mitral cells to acetophenone and the 12 other odorants was significantly increased (*Figure 3D*, Two-sample Kolmogorov-Smirnov test: $D_{4116,1639}$ = 0.24, p<0.01, and *Figure 3—figure supplement 1*).

Finally, we analyzed the trial-to-trial variability of mitral cell responses following the repeated delivery of the same odorant. Mice were presented with the same odorant 4 times (average inter-trial interval ~10 min), and the presentation of each odorant was interleaved with other odorants to avoid habituation. In littermate controls, 56% of responsive mitral cells responded on only one out of 4 trials, 20% of cells responded twice, 10% three times, and 14% of cells responded on all 4 out of 4 trials (*Figure 3—figure supplement 1*). In M71 transgenic mice, the fraction of neurons responding on all 4 out of 4 repeat presentations of the same odorant was reduced from 14% in controls to 2.9% in M71 transgenic mice. Furthermore, we calculated the Pearson correlation coefficients of the activity patterns after odor onset. We found that for acetophenone and the 12 other odorants, the mean correlation of response patterns to individual exposures of the same odorant was reduced in M71 transgenic mice compared to controls (controls: mean across 13 odorants = 0.75 ± 0.11; M71: mean = 0.25 ± 0.17, *Figure 3E and F*).

Taken together, these data suggest that the neural circuits of the olfactory bulb of M71 transgenic mice can greatly amplify weak odor-evoked signals while suppressing overly strong signals. Such amplification may explain how M71 transgenic mice can still detect and discriminate most odorants. However, an increase in the trial-to-trial variability of mitral cell responses will degrade the fidelity of the odor representation, and may underlie the impairments in odor discrimination that M71 transgenic mice exhibit with more challenging assays.

## Intrinsic and odor-evoked mitral cell activity in M71 transgenic mice

Our calcium imaging experiments provide information about the patterns of odor-evoked activity in large ensembles of mitral cells. We next sought to obtain more detailed information about the network mechanisms underlying the normalization of odor-evoked mitral cell activity, using whole cell recordings from mitral and tufted cells (MTCs) in awake head-fixed mice. First, we characterized the intrinsic properties of MTCs, including resting membrane potentials, input resistance, membrane time constant tau, and baseline firing rates. These biophysical properties of MTCs were, on average, similar in M71 transgenic mice (n = 6 cells from 6 mice) and controls (n = 7 cells from 5 mice, *Figure 4A–G*). Interestingly, however, we observed that MTCs in M71 transgenic mice appeared to be less heterogeneous compared to wild-type, in particular for baseline firing rate (Control: 5.8 ± 6.2 Hz, M71: 3.0 ± 1.0 Hz, p=0.003 Bartlett test) and theta modulation strength (Control: 0.4 ± 0.4

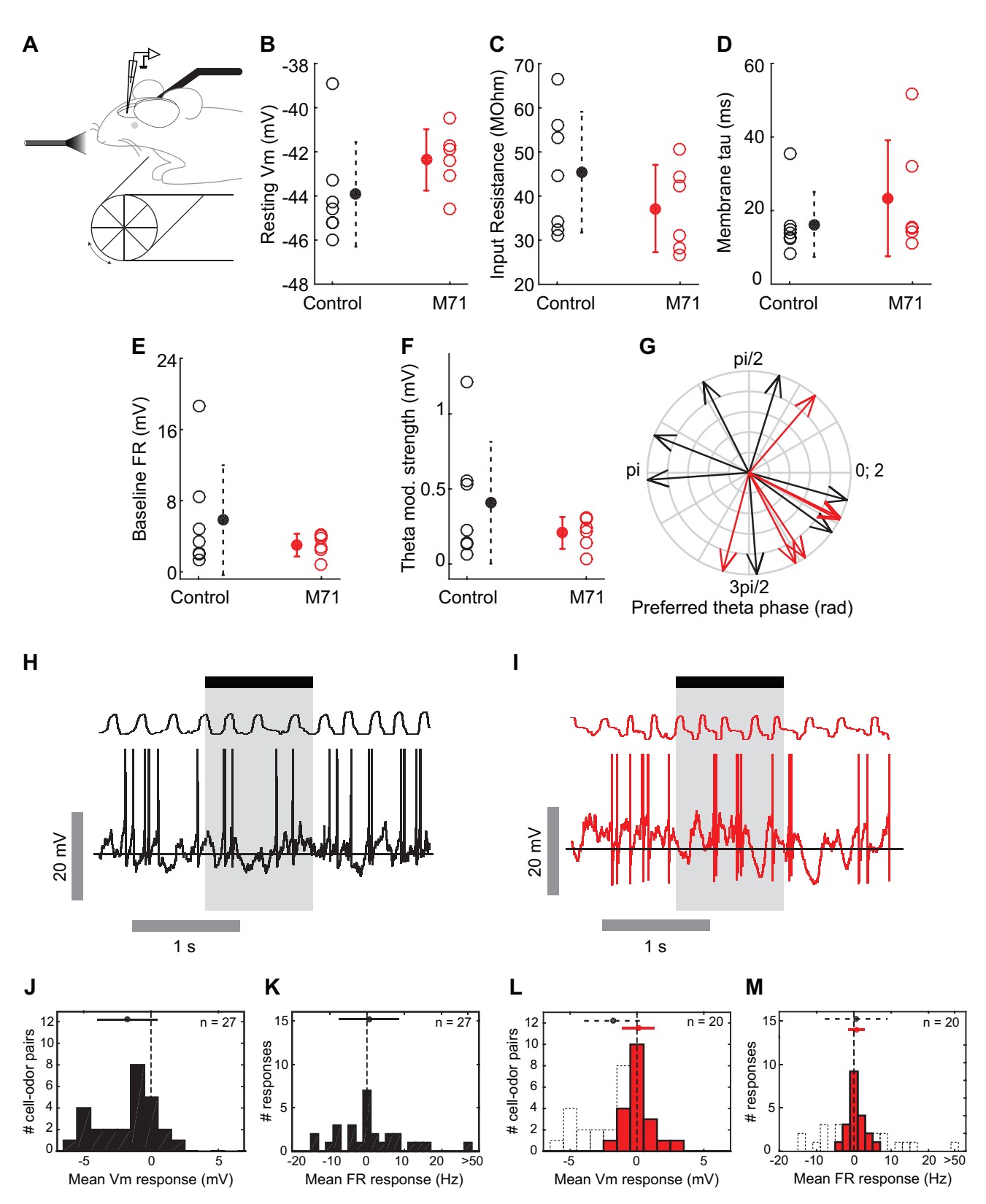

**Figure 4.** Intrinsic and odor-evoked mitral cell activity in M71 transgenic mice. (A–M) In vivo whole cell recordings in awake mice: comparison of physiological properties of mitral cells in control (black, n = 7) and M71 transgenic mice (red, n = 6). (A) Schematic of the experimental configuration. (B)
*Figure 4 continued on next page*

*Figure 4 continued*

Resting membrane potential (mV), (**C**) input resistance (MΩ), (**D**) membrane time constant tau (ms), (**E**) baseline firing rate (Hz), (**F**) strength of modulation of baseline Vm by the sniff cycle (theta coupling) (mV), and (**G**) phase-preference of baseline Vm within the sniff cycle (rad). (**H, I**) Example trace showing single 1 s odor presentation (shaded area) during mitral cell recordings from control (**H**) and M71 transgenic (**I**) mice. (**J, K**) Histograms of mean odor-evoked membrane potential (**J**) and firing rate (**K**) responses in control cells to non-acetophenone odors, n = 27 cell-odor pairs from 7 cells. (**L, M**) Histograms of mean odor-evoked membrane potential (**L**) and firing rate (**M**) responses in M71 transgenic cells, n = 20 cell-odor pairs from 6 cells. Black dotted lines in (**L**) and (**M**) show corresponding data from controls.

The following figure supplement is available for figure 4:

**Figure supplement 1.** Patch clamp mitral cell recordings in anesthetized mice reveal increased acetophenone-evoked inhibition in M71 transgenic mice.

mV, M71: $0.2 \pm 0.1$ mV, p=0.01 Bartlett test), which might reflect their more homogeneous olfactory inputs and thus developmental history (*Angelo et al., 2012*).

Next, we sought insight into how responses to odorants other than acetophenone are able to evoke largely normal levels of mitral cell activity, despite the dramatic reduction in the expression of endogenous odorant receptors in M71 transgenic mice. We measured evoked MTC response profiles to a one-second pulse of 4 non-acetophenone stimuli (3 monomolecular odorants - hexanone, heptanal, ethyl acetate - and a mixture of isoamyl acetate, 2-nonanone and cyclohexanol, at a concentration of 1% of absolute vapor pressure). As previously reported (*Cury and Uchida, 2010*; *Kollo et al., 2014*; *Shusterman et al., 2011*), MTC activity in these awake, head-fixed mice was modulated by odor in a diverse manner, with prominent excitatory as well as inhibitory responses (*Figure 4H,J and K*). On average, odor exposure resulted in a moderate hyperpolarization (mean $\Delta$Vm = -1.8 mV, SD = 2.1 mV, 27 cell/odor pairs), and a small increase in the firing rate (mean $\Delta$ firing rate = 2.2 Hz, SD = 13.3 Hz, 27 cell/odor pairs). Consistent with our imaging data, we found that overall firing rate distributions to the same 4 stimuli were more compact in M71 transgenic mice; mean changes in firing rate were indistinguishable from controls (mean $\Delta$ firing rate$_{M71}$ = 0.75 Hz, SD = 2.2 Hz, 20 cell/odor pairs, p=0.67, Rank-sum, *Figure 4M*), yet the fraction of excitatory responses was similar (control: 32%, M71: 25%). Odor presentation generally resulted in both excitatory and inhibitory responses, with only a small change in the average membrane potential (mean $\Delta$Vm = 0.10 mV, SD = 1.1mV, 20 cell/odor pairs; *Figure 4I,J and M*). However, both excitatory and inhibitory responses were generally weaker ($\Delta|$Vm$|_{M71}$ = 0.5+[-0.3 0.8] mV, $\Delta|$Vm$|_{cntrl}$ = 1.3+[-0.6 2.5] mV, p=0.006, Wilcoxon rank sum; $\Delta |$firing rate$|_{M71}$ = 0.8+[-0.3 1.6]yy Hz, $\Delta|$firing rate$|_{cntrl}$ = 6.0+[-3.0 6.2] Hz, p<0.001, Wilcoxon rank sum, median +[lower quartile, upper quartile]), and strong excitatory responses notably absent in M71 transgenic mice (*Figure 4M*). Most prominently, inhibitory responses were substantially reduced compared to littermate controls (p=0.003, Rank-sum, *Figure 4L*).

Taken together, calcium imaging experiments and in vivo whole cell recordings reveal overall surprisingly normal mitral cell odor responses in M71 transgenic mice, despite massive changes in odor-evoked sensory input. Importantly, however, responses to odorants other than acetophenone result in slightly weaker, more variable responses and in particular - as apparent from the subthreshold analysis - substantially less hyperpolarizing responses. Finally, as in the awake case, in anaesthetized M71 transgenic mice responses to non-acetophenone odors were weaker than in controls (*Figure 4—figure supplement 1*).

## Acetophenone induces strong and prolonged inhibition and a massive increase in theta coupling in M71 transgenic mice

In contrast to all our other test odorants, acetophenone strongly activates the vast majority of sensory neurons, resulting in pervasive glomerular activity in M71 transgenic mice (*Fleischmann et al., 2008*). Despite this widespread glomerular activation, our calcium imaging experiments have demonstrated that acetophenone activates similar numbers of mitral cells in both control and M71 transgenic mice. One mechanism behind this apparent normalization could be inhibition that is increased concomitantly with the massively increased excitatory input.

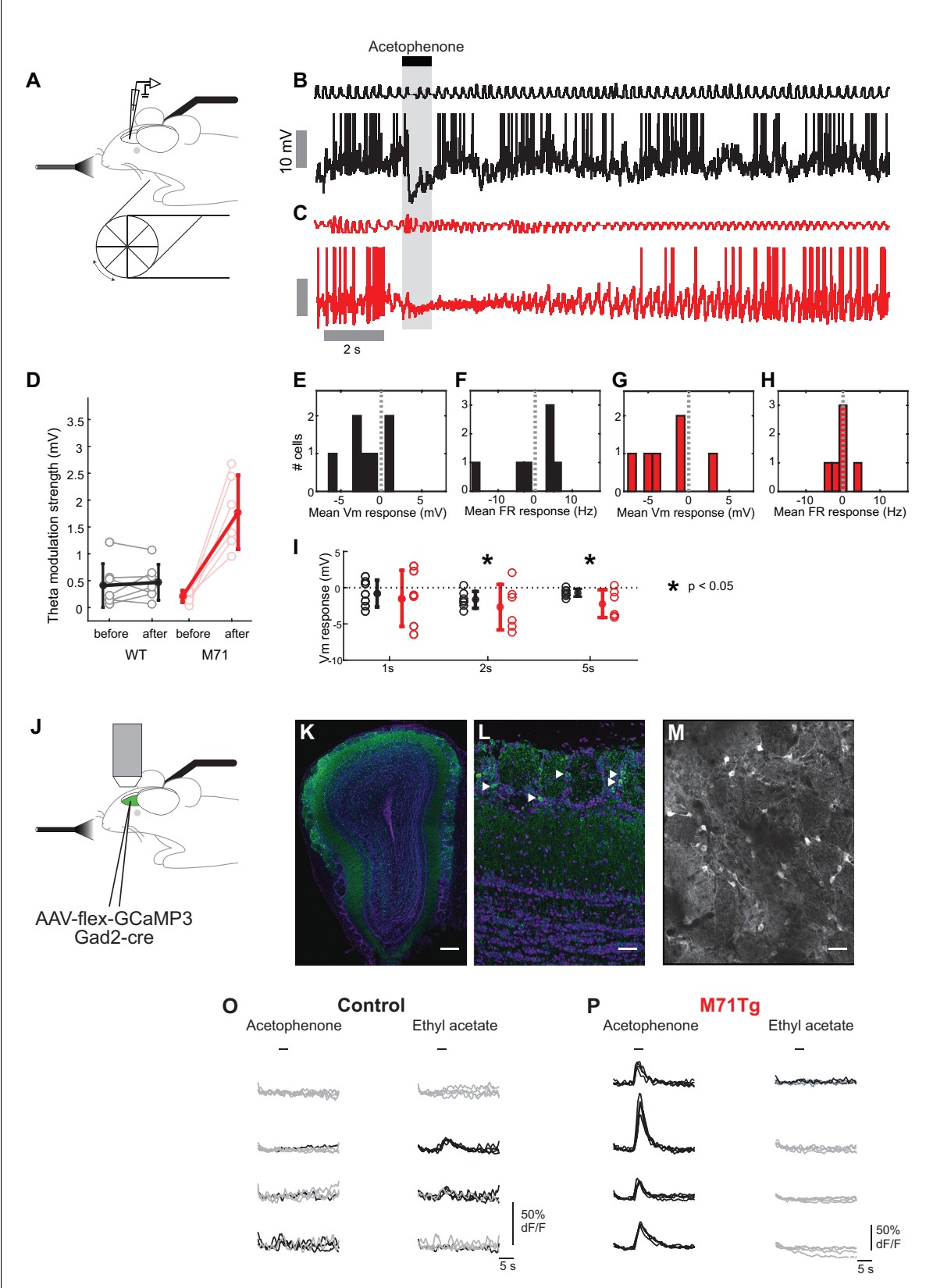

**Figure 5.** Increased acetophenone-evoked inhibition and theta coupling in M71 transgenic mice. (A–H) In vivo whole cell recordings in awake mice. (A) Schematic of the experimental configuration. (B, C) Example trace of a 1% acetophenone presentation to a mitral cell in a control (B, black) and M71 *Figure 5 continued on next page*

*Figure 5 continued*

transgenic (**C**, red) mouse. Note the differences in the duration of inhibition, and the great amplification in theta coupling of the M71 transgenic cell after response offset. (**D**) Quantification of strength of theta coupling before and after acetophenone presentation for control (black) and M71 transgenic (red) cells. (**E–H**) Histograms of mean 1 s odor-evoked membrane potential and firing rate responses to acetophenone presentation in control (**E**, **F**) and M71 transgenic (**G**, **H**) mice. (**I**) Comparison of control and M71 transgenic mean Vm responses calculated over different time windows: 1 s, 2 s and 5 s from the first inhalation post odor-onset. (**J–P**) Two-photon in vivo imaging of periglomerular cell activity in anesthetized mice. (**J**) Schematic of the experimental configuration. (**K**) Expression of GCaMP3 (in green) in inhibitory neurons after injection of conditional AAV-flex-GCaMP3 into the olfactory bulb of a *Gad2-Cre* transgenic mouse. Purple: nuclear counterstain. Scale bar = 100 μM. (**L**) Higher magnification of periglomerular (PG) cells (examples indicated by white arrowheads) expressing GCaMP3. Scale bar = 20 μM. (**M**) Two-photon micrograph showing GCaMP3 expression in PG cells of a single imaging site. Scale bar = 20 μM. (**O**, **P**) Example traces of the responses of 4 PG cells to acetophenone and ethyl acetate in control (**O**) and M71 transgenic (**P**) mice. Traces represent responses to 4 individual odorant exposures, non-responsive trials are shown in grey, responsive trials in black. Note the difference in the scale of the y-axis between genotypes.

The following figure supplement is available for figure 5:

**Figure supplement 1.** Individual acetophenone response traces.

To directly test this prediction we also examined acetophenone-evoked mitral cell activity in M71 transgenic mice using whole cell patch-clamp recordings. Strikingly, in M71 transgenic mice, acetophenone exposure caused a massive and prolonged increase in theta modulation of the membrane potential (*Figure 5C and D*). Phasic odor responses, however, were highly similar in control and M71 transgenic mice: mean firing rate change induced by acetophenone was again not significantly different between M71 transgenic mice and controls (control: -0.2 ± 7.5 Hz, M71: -0.5 ± 2.7 Hz, p=0.89, t-test, *Figure 5F and H*), consistent with results obtained in calcium imaging experiments. However, acetophenone generally induced prolonged hyperpolarizations in M71 transgenic mice, whereas responses were more transient in littermate controls (responses over 5 s window: control: -0.7 ± 0.5 mV, M71: -2.2 ± 1.9 mV, p<0.05, 1-tailed t-test. *Figure 5E,G and I*, and *Figure 5—figure supplement 1*).

Thus, while supra-threshold responses in mitral cells are highly similar between control and M71 transgenic mice, whole-cell recordings in awake animals reveal a potential source of this normalization: hyperpolarizing, inhibitory responses are increased for the M71 receptor ligand acetophenone but reduced for other odorants compared to control animals (cf. *Figure 4J and L* and *Figure 5I*). These alterations in inhibition were not a consequence of altered sampling behavior e.g. due to anxiety (*Glinka et al., 2012*) or other behavioral state changes, as whole-cell recordings in anaesthetized mice showed a similarly profound and selective increase in inhibitory responses to acetophenone exposure (*Figure 4—figure supplement 1*).

Previous work indicated that such slow, odor-evoked phasic inhibition likely originates in the glomerular layer (*Fukunaga et al., 2014*), and the position and connectivity of PG cells make them prime candidates to mediate both presynaptic and feedforward inhibition in response to acetophenone in M71 transgenic mice. Therefore, we performed two-photon imaging experiments in mice engineered to selectively express GCaMP3 in PG cells. Selectivity was achieved by injecting Cre-dependent AAV (AAV5.hSynap.Flex.GCaMP3.WPRE.SV40) into the olfactory bulbs of either M71 transgenic mice that also carried a *Gad2-Cre* transgene (*Taniguchi et al., 2011*), or littermate controls expressing the *Gad2-Cre* transgene only. AAV injections resulted in the labeling of large numbers of GABA-positive interneurons in the glomerular layer, with extensive processes extending into individual glomeruli (*Figure 5K–M*). In control mice, only a fraction of PG cells displayed responses to either acetophenone or ethyl acetate (acetophenone: 10.9%; ethyl acetate: 12.1%, *Figure 5O* and *Figure 6G*). The magnitudes of these odor-evoked responses were small, with more than 80% of peak ΔF/F values below 10% (data not shown). Unlike mitral cells, where responses in M71 transgenic mice and their littermate controls were often indistinguishable, odor responses were strikingly different in PG cells. Exposure of M71 transgenic mice to acetophenone, even at the lowest concentration (0.01% vol./vol.), resulted in pervasive, strong and persistent activity in over 48% of PG cells, significantly higher than in littermate controls (*Figure 5P* and *Figure 6H*, Rank-sum test $n_{co}$ = 10, $n_{M71\ transgenic}$ = 9, U = 83, p<0.01). Furthermore, the magnitude and duration of acetophenone-evoked PG cell activity was significantly increased in M71 transgenic mice compared to controls (data not shown). In contrast to these robust and pervasive responses to acetophenone, ethyl

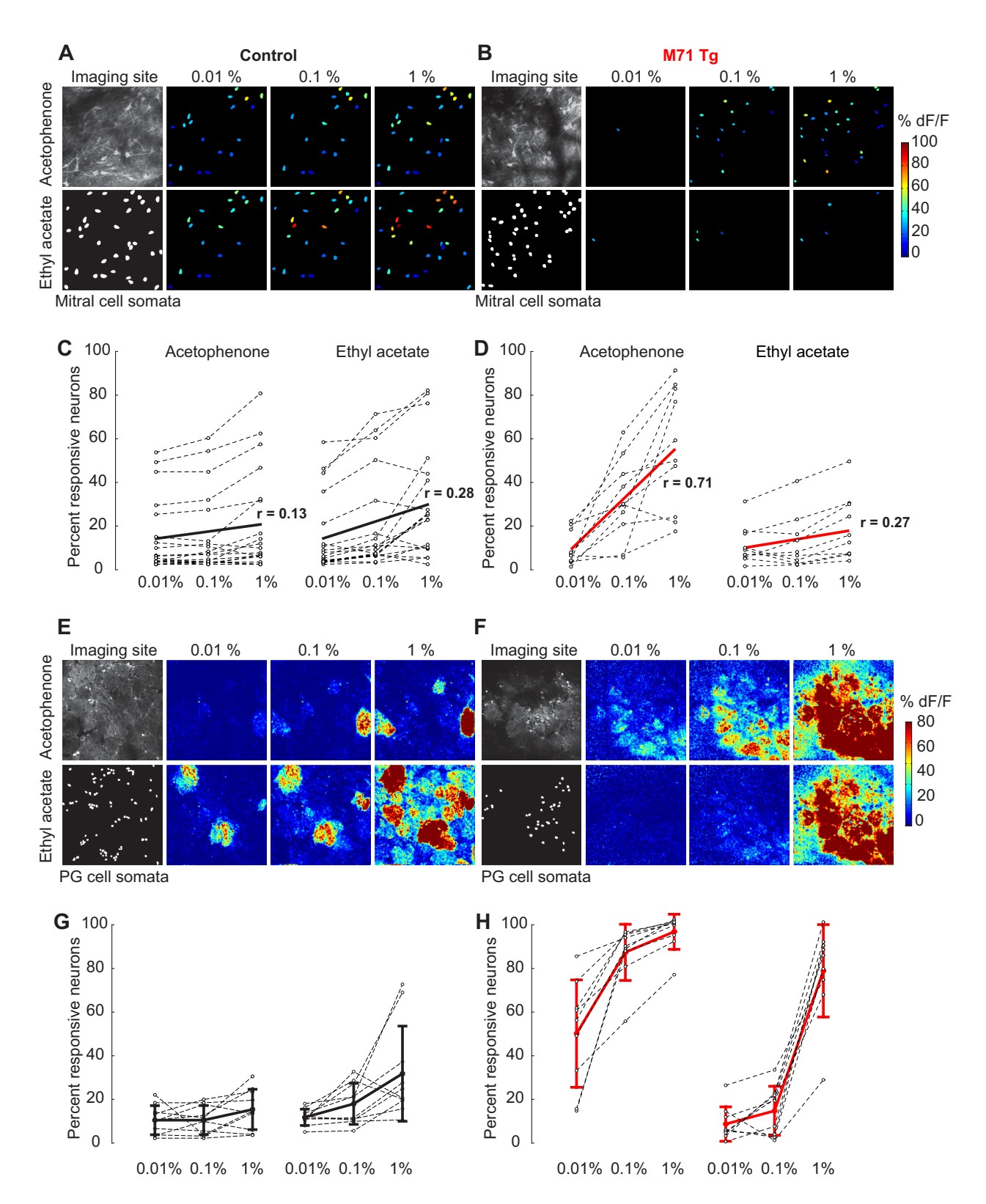

**Figure 6.** The olfactory bulb excitation/inhibition balance in M71 transgenic mice breaks down at high acetophenone concentrations. (**A–D**) Two-photon in vivo imaging of mitral cell odor responses in anesthetized mice. (**A, B**) Representative maps of odor-evoked mitral cell activity elicited by

*Figure 6 continued on next page*

*Figure 6 continued*

acetophenone and ethyl acetate at increasing odorant concentrations in a control (**A**) and M71 transgenic (**B**) mouse. Weak to strong responses are color-coded in blue to red. (**C, D**) Mean fraction of mitral cells that respond to acetophenone and ethyl acetate at increasing odorant concentrations in control (**C**, black) and M71 transgenic (**D**, red) mice. Grey circles represent the fraction of responsive cells of a single imaging site. Controls: 19 imaging sites in 8 mice, n (median number of cells per site) = 57; M71 transgenics: 10 imaging sites in 4 mice, n = 28. red line: linear fit onto concentration. r: coefficient of correlation. (**E–H**) Two-photon in vivo imaging of periglomerular cell activity in anesthetized mice. (**E, F**) Representative maps of odor-evoked periglomerular (PG) cell activity elicited by acetophenone and ethyl acetate at increasing odorant concentrations in a control (**E**) and M71 transgenic (**F**) mouse. Note that the heatmaps predominantly reflect dendritic responses of PG cells in glomeruli. (**G, H**) Mean fraction of PG cells that respond to acetophenone and ethyl acetate at increasing odorant concentration in control (**G**, black) and M71 transgenic (**H**, red) mice. Grey circles represent the fraction of responsive cells of a single imaging site. Controls: 10 imaging sites in 5 mice, n (median number of cells per site) = 46; M71 transgenics: 9 imaging sites in 5 mice, n = 51. Error bars = SD.

The following figure supplement is available for figure 6:

**Figure supplement 1.** Response magnitudes and durations strongly increase with increasing acetophenone concentrations in M71 transgenic mice.

acetate elicited PG cell activity in only a small population of neurons (<10%, *Figure 5P* and *Figure 6H*), and response magnitudes were consistently below 10% peak ΔF/F values (data not shown). These responses tended to be fewer and with smaller ΔF/F values than those observed in littermate controls, but this observation did not reach statistical significance. Taken together, electrophysiology and imaging experiments indicate that pervasive and strong glomerular excitation is balanced by similarly pervasive and strong periglomerular inhibition to normalize olfactory bulb mitral cell output.

## Inhibition-mediated normalization breaks down at high acetophenone concentrations

We next attempted to upset this balance of mitral cell excitation and PG cell inhibition by increasing odorant concentration. As mentioned above, acetophenone and ethyl acetate at low concentrations (0.01% vol./vol.) activate approximately 10% of mitral cells in both M71 transgenic mice and controls (*Figure 6C and D*). In control mice, a 10- and 100-fold increase in acetophenone or ethyl acetate concentration only caused a modest increase in mitral cell activity: about 15% of mitral cells responded to either odorant at 0.1% dilutions, and ~20% of mitral cells responded at 1% dilutions (*Figure 6A and C*). Response magnitudes and durations increased slightly while trial-to-trial variability decreased with increasing odor concentrations (*Figure 6—figure supplement 1*).

In marked contrast to controls, increasing concentrations of acetophenone in M71 transgenic mice dramatically increased the fraction of responsive mitral cells: acetophenone at 0.1% activated over 28% of mitral cells (mean = 28.7% ± 17.1%), while over 50% (mean = 50.1% ± 25.8%) of mitral cells responded at 1% acetophenone (*Figure 6B and D*). The number of responsive neurons was highly correlated to acetophenone concentration (n = 10, Pearson's correlation coefficient = 0.71), indicating that the strong dependence of acetophenone responses on concentration is consistently observed across all imaging sites. Furthermore, although response magnitudes sharply increased with increasing acetophenone concentrations, response durations were reduced (*Figure 6—figure supplement 1*). The striking sensitivity to acetophenone concentration was not observed for ethyl acetate. A ten-fold increase in ethyl acetate concentration resulted in only a ~1.5-fold increase in the fraction of responsive mitral cells, and a hungred-fold increase in ethyl acetate activated ~2 times more cells, essentially identical to what we observed in controls (*Figure 6B and D*). We did not observe robust concentration-dependent changes in the distributions of peak ΔF/F values and response durations to ethyl acetate (*Figure 6—figure supplement 1*). To quantify these differences, we calculated the difference in concentration-driven linear change between stimuli ($\Delta LC = LC_{acetophenone} - LC_{ethyl\ acetate}$) in control and M71 transgenic mice. A $\Delta LC$ of 0 indicates no difference in the effect of concentration on the response to the two stimuli, while a positive $\Delta LC$ indicates the greater linear change in acetophenone-evoked response density with increasing concentration. We found no difference in linear change between ethyl acetate and acetophenone responses for controls ($\Delta LC = -0.04$), indicating that control mice responded similarly to increasing concentrations of these 2 stimuli. In contrast, we found a positive $\Delta LC$ in M71 transgenic mice ($\Delta LC = 0.18$), highlighting that the density of neural responses to acetophenone is strongly modulated by concentration in these mice.

Earlier we suggested that PG cell-mediated feedforward inhibition plays an important role in normalizing olfactory bulb output (*Figure 5*). We therefore next asked how increasing odorant concentration affects the activity of PG cells in both M71 transgenic mice and controls. While PG cells were already quite responsive to low acetophenone concentrations in M71 transgenic mice, increasing acetophenone concentrations dramatically increased PG cell activity in these mice (*Figure 6F and H*): 0.1% activated over 85% (mean 85.1% ± 13%) of PG cells in M71 transgenic mice compared to only 10.8% ( ± 6.6%) observed in controls. A further ten-fold increase in concentration (1%) increased the fraction of responsive neurons to 94% ( ± 8%) in M71 transgenic mice. Increasing acetophenone concentration also markedly increased response magnitudes and durations, which was not observed in controls (data not shown). Interestingly, exposure of M71 transgenic mice to intermediate concentrations of ethyl acetate at (0.1%) activated only 14.6% ( ± 11%) of PG cells, similar to the 18.2% ( ± 9.3%) responsive PG cells observed in littermate controls (Mann-Whitney test $n_{co}$ = 10, $n_{M71\ transgenic}$ = 9, U = 35, p=0.45). Interestingly, however, a hundred-fold increase in ethyl acetate concentration (1%) activated 77.9% ( ± 20%) of PG cells, more than double that observed in controls. Although ethyl acetate has previously been reported not to activate the M71 receptor (*Bozza et al., 2002*), these data suggest that ethyl acetate may in fact be a weak M71 receptor agonist that activates M71 at high concentrations. This speculation is supported by the observation that while odorants activate segregated, glomerulus-specific clusters of PG cells in wild-type mice, PG cell responses to both acetophenone and ethyl acetate were pervasive and not restricted to individual glomeruli in M71 transgenic mice, consistent with the pervasive glomerular innervation by M71-expressing sensory inputs (*Figure 6E and F*).

Taken together, these data demonstrate that the pervasive glomerular activity elicited by acetophenone in M71 transgenic mice results in strong, stimulus-specific PG cell activation, even at low concentrations. At higher concentrations, acetophenone-evoked PG cell activity appeared to saturate in M71 transgenic mice while the proportion of responsive mitral cells dramatically increased. These data therefore suggest that at high acetophenone concentrations, periglomerular inhibition fails to counterbalance excitation.

## Discussion

We have characterized the transformation of odor-evoked activity in the olfactory bulb of M71 transgenic mice with a 'monoclonal nose', in which glomerular input patterns are massively perturbed. Calcium imaging experiments reveal that olfactory bulb mitral cell output in M71 transgenic mice is surprisingly similar to that observed wild-type mice. The olfactory bulb of M71 transgenic mice thus provides a powerful model system with which to explore neural mechanisms of signal normalization. Cell-specific calcium imaging together with in vivo patch-clamp recordings identify feedforward periglomerular inhibition as a major candidate neural circuit mechanism for signal amplification and suppression. Our results thus highlight the capacity of the olfactory bulb to extract meaningful information from degraded sensory input.

### The ability of M71 transgenic mice to detect acetophenone is task-dependent

Our previous behavioral analyses of M71 transgenic mice using a go/no go operant conditioning task indicated that M71 transgenic mice could readily detect and discriminate most odorants. In contrast, M71 transgenic mice failed to discriminate the M71 receptor ligand acetophenone from air. We have replicated these initial findings with an independent cohort of mice. In total, 15 M71 transgenic mice were tested in this task, and all 15 mice failed to reach a correct lick ratio of above 75%. Why do M71 transgenic mice fail this test? We previously hypothesized that pervasive glomerular activation elicits inhibition at multiple stations along the olfactory pathway, which could entirely suppress acetopheone-evoked neural activity. However, our calcium imaging and electrophysiological recordings do not support this hypothesis. Instead, we observe that acetophenone-evoked suprathreshold mitral cell activity in M71 transgenic mice is surprisingly similar to that observed in wild-type mice: the fraction of acetophenone-responsive neurons, and acetophenone-induced changes in mitral cell firing rates are indistinguishable from controls. Furthermore, alterations in mitral cell response magnitude, duration, and changes in the dynamic range of mitral cell responses do not provide a simple explanation for the observed behavioral deficit. The most striking acetophenone-

specific effect we observe is a massive amplification of theta oscillations. A single one second puff of acetophenone elicits strong theta oscillations that last for at least 20 s. The behavioral consequences of this oscillatory activity are unknown, but it is possible that these prolonged network perturbations interfere with the discrimination of acetophenone from clean air in the go/no go operant conditioning task. Alternatively, we cannot exclude that our clean air stimulus contains contaminating traces of acetophenone, and that acetophenone at extremely low concentrations is sufficient to elicit neural activity in M71 transgenic mice that makes it indistinguishable from the acetophenone odor puff. This would suggest that the observed behavioral deficit results from the inability of M71 transgenic mice to discriminate between different acetophenone concentrations, rather than from the inability to detect it. Finally, strong oscillatory network activity may precede an epileptic state (*Nguyen and Ryba, 2012*), which could obstruct odor discrimination.

In contrast to their inability to discriminate acetophenone from air in the go/no go operant conditioning task, M71 transgenic mice adapt their spontaneous sniffing behavior in response to acetophenone exposure, suggesting that they can indeed detect acetophenone in this test. Spontaneous sniff adaptation does not require that mice accurately discriminate between successive presentations of two different stimuli. Therefore, while acetophenone-induced network perturbations may interfere with odor discrimination, mice may be able to detect and recognize acetophenone as a previously encountered stimulus in this test. Additionally, the heightened level of anxiety in M71 transgenic mice (*Glinka et al., 2012*) might amplify sniff responses to novel stimuli including to non-olfactory components of the stimulus. Another important difference between the two behavioral tasks is that unlike spontaneous sniff adaptation, operant conditioning requires extensive training. It is possible that training, which is performed with non-acetophenone odorants, may shape the processing of odor-evoked activity, and that such plastic changes may underlie the task dependency of the behavioral deficit.

## Suppression of pervasive glomerular activity in M71 transgenic mice

Changes in the expression of odorant receptor genes in M71 transgenic mice have two major consequences for odor-evoked glomerular activity. First, exposure to the M71 receptor ligand acetophenone activates the vast majority of sensory neurons and elicits pervasive glomerular activity. Second, the number of sensory neurons responsive to most odorants, i.e. those that do not activate the M71 receptor, is strikingly reduced, resulting in glomerular activity that is below the detection threshold of in vivo synapto-pHluorin imaging experiments (*Fleischmann et al., 2008*). Despite these massive perturbations in odor-evoked olfactory bulb input, both our calcium imaging experiments and electrophysiological recordings reveal that mitral cell responses to acetophenone and non-acetophenone odorants are highly similar. Thus, M71 transgenic mice provide an exaggerated genetic setting in which we have examined neural circuit mechanisms for generating normalized sensory output for downstream targets that direct olfactory behaviors.

The fraction of acetophenone-responsive mitral cells as well as acetophenone-evoked changes in mitral cell firing rates are indistinguishable between M71 transgenic mice and controls. This observation immediately suggests that powerful inhibitory mechanisms must exist within the olfactory bulb to prevent massive excitation evoked by pervasive inputs to the glomerular layer. We now show that PG cells are pervasively and robustly activated by acetophenone in M71 transgenic mice. Furthermore, mitral cell patch-clamp recordings reveal that acetophenone evokes strong and prolonged phasic inhibition in M71 transgenic mitral cells, reminiscent of PGC-mediated feedforward inhibition described previously (*Fukunaga et al., 2014*). This feedforward inhibitory activity is likely to provide an effective mechanism to transform pervasive glomerular activity into sparse mitral cell responses.

Interneuron populations in the deeper layers of the olfactory bulb could further modify neural activity evoked by acetophenone. Candidates include superficial dopaminergic interneurons that have previously been suggested to mediate signal normalization (*Banerjee et al., 2015*), as well as parvalbumin-expressing interneurons, which reside in the external plexiform layer and receive direct input from widely distributed mitral cells (*Kato et al., 2013*; *Miyamichi et al., 2013*). In addition, deep layer granule cells, which form dendrodendritic synapses on the lateral dendrites of nearby mitral cells can modulate neighboring mitral cell output by means of powerful feedback and feedforward inhibition (*Abraham et al., 2010*; *Isaacson and Strowbridge, 1998*; *Jahr and Nicoll, 1982*; *Margrie et al., 2001*; *Rall et al., 1966*). Altogether, these interglomerular inhibitory networks can normalize response magnitudes across a range of input intensities and enhance contrast between

patterns of odor-evoked glomerular activity, and feedforward inhibition from primary onto secondary olfactory neurons represents an olfactory circuit function that appears highly conserved in evolution (*Olsen and Wilson, 2008*; *Zhu et al., 2013*). However, we found that increasing acetophenone concentration dramatically increased the fraction of responsive mitral cells, resulting in dense neural odor representations similar to the dense patterns of acetophenone-evoked glomerular activity. Thus, at high acetophenone concentrations, excitation may override inhibition, exposing the limits of olfactory bulb circuit mechanisms to normalize glomerular activity.

## The amplification of weak sensory inputs in the olfactory bulb of M71 transgenic mice

We observed that all odorants tested elicit sparse and unique, overlapping patterns of mitral cell activity in M71 transgenic mice. Strikingly, the fraction of responsive neurons to a panel of 13 odorants (including acetophenone) was not significantly different from what was observed in littermate controls. While individual ligand-receptor interactions remain incompletely characterized, calcium imaging experiments suggest that most odorants do not activate M71-expressing olfactory sensory neurons, or do so only at high odorant concentrations (*Bozza et al., 2002*; *Fleischmann et al., 2008*). Thus, our data provide a striking example of signal amplification by olfactory bulb neural circuits.

Whole cell recordings of mitral cells reveal that odorants commonly evoke phasic inhibition in wild-type mice (as described previously e.g *Fukunaga et al., 2014*; *Margrie et al., 2001*), and that this inhibition in response to odorants other than acetophenone is significantly reduced in M71 transgenic mice. This result suggests that olfactory bulb inputs ordinarily evoke non-specific inhibition that can only be overcome by strong and specific input. Such inhibition coupled with specific excitation can increase signal-to-noise ratios and enhance contrast (*Cleland and Sethupathy, 2006*). By contrast, in the M71 transgenic mice, non-acetophenone inputs are too weak to recruit inhibition, allowing these weak inputs to evoke responses. This may at least partially explain why responses are prolonged, and why trial-to-trial variability of responses is increased in these animals. However, weak signals may also be actively amplified. For example, electrical coupling electrical coupling between mitral cells connected to the same glomerulus, and self-excitation of intraglomerular mitral cell assemblies, can further facilitate the detection of weak odor signals (*Christie et al., 2005*; *Isaacson, 1999*; *Margrie et al., 2001*; *Murphy et al., 2005*; *Schoppa and Westbrook, 2001*). However, a multi-synaptic pathway involving olfactory bulb tufted cells may also directly amplify the output of mitral cells in response to weak inputs (*De Saint Jan et al., 2009*; *Fukunaga et al., 2012*; *Gire et al., 2012*; *Najac et al., 2011*). These different neural mechanisms are likely to cooperate in improving the ability of M71 transgenic mice to detect odorants that do not activate the M71 receptor. Interestingly, however, we observed that the patterns of mitral cell activity are more variable across multiple odorant presentations. The fraction of mitral cells responding reliably to the same stimulus is significantly reduced in M71 transgenic mice, and the variation of the average fraction of responsive neurons for a given trial is increased. Odorants will therefore activate more variable ensembles of cells. One important source of variability in neural responses to sensory stimuli is noise, and neural circuit mechanisms to reduce variability due to noise often rely on averaging signals from neurons carrying redundant information (*Faisal et al., 2008*). Large numbers of sensory neurons expressing the same odorant receptor and converging onto only two glomeruli in the olfactory bulb provide a striking example of this principle. In M71 transgenic mice, the number of sensory neurons expressing a given odorant receptor are strongly reduced, thus limiting the power of averaging to reduce variability. We speculate that this increased variability of a neural odor representation will affect the accuracy of odor discrimination in M71 transgenic mice, a model consistent with the olfactory discrimination deficits observed for difficult to discriminate odorant mixtures.

In conclusion, we report a number of ways in which the olfactory bulb can modify substantially altered primary inputs to generate meaningful odor representations. Amplification of weak signals and suppression of strong, pervasive input patterns are likely to be crucial under normal circumstances, for example by allowing the system to tune to odors with considerable variations in vapor pressures.

# Materials and methods

## Mice

Adult (6–10 week-old) mice on a mixed 129SvEv; C57BL/6 genetic background were used for all experiments. *Omp-ires-tTA* and *tet$_o$-M71-ires-lacZ* mouse lines were bred to generate hemizygous *Omp-ires-tTA /tet$_o$-M71-ires-lacZ* transgenic mice (referred to as M71 transgenic mice). Wild-type and *Omp-ires-tTA* heterozygous littermates were used as controls. To generate compound *Omp-ires-tTA; tet$_o$-M71-ires-lacZ; Gad2-Cre* transgenic mice (*Taniguchi et al., 2011*), *Omp-ires-tTA / tet$_o$-M71-ires-lacZ* females were bred with homozygous *Gad2-Cre* males. *Gad2-Cre* littermates were used as controls. All experiments were performed according to Columbia University, College de France, and the Francis Crick Institute institutional animal care guidelines.

## Behavior

Go/no go operant conditioning experiments were performed in a liquid dilution, eight channel olfactometer (Knosys, Lutz, Florida) as described previously (*Bodyak and Slotnick, 1999*; *Fleischmann et al., 2008*). Briefly, mice were water-restricted (1–1.5 ml water/day) and maintained on a reverse 12 hr light/dark cycle. Initial training was performed with ethyl acetate, citronellol, and carvone. All odorants were used at 1% vol./vol. dilution in mineral oil. Individual experiments consisted of at least 200 trials and typically lasted for ~30 min. Individual trials consisted of a 2 s odor sampling period, followed by an inter-trial interval of at least 4 s. The median time from the end of one odor presentation (closing of the odor valve) to the beginning of the next was $6.3 \pm 0.5$ s (mean and SD across 7 animals, 300 trials each). Discrimination accuracy was calculated as the percent correct licks during a two second interval following valve opening for blocks of 20 trails each. Behavioral data were analyzed in R by fitting a linear mixed-effects model to test the effect of genotype on the fraction of correct licks (fraction correct lick ~ genotype * block + 1 | mouse Id / block).

Sniff behavior was measured in head-fixed passive mice using a fast mass flow sensor (FBAM200DU, Sensortechnics, Puchheim, Germany) externally placed in close proximity to the left nostril. Baseline sniff frequencies for each trial were calculated by taking the inverse of the mean inter-sniff interval (time between successive inhalation peaks) during the 2 s prior to odor period. Responses were calculated by subtracting the baseline sniff frequency from the sniff frequency measured similarly for all sniff cycles beginning and ending within the odor period. Inter-trial interval was 10 s and 8 odor stimuli were presented in a fixed order (1% acetophenone, ethyl acetate, 0.5% acetophenone, mixture 1, 0.1% acetophenone, hexanone, 0.05% acetophenone, heptanal). This block was repeated at least 3 times. The sequence of odors was randomized between animals ensuring alternating acetophenone and non-acetophenone odors. No blank controls were presented. For analysis, sniff responses were analyzed for the first and second/third acetophenone presentation irrespective of concentration.

## Rabies-GCaMP3 virus Injections

Deletion-mutant rabies virus was generated as described in *Wickersham et al. (2010)*. Mice were anaesthetized with ketamine/xylazine (100 mg/kg / 10 mg/kg, Sigma Aldrich) and body temperature was maintained at 37°C using a feedback-controlled heating pad (Fine Science Tools). The scalp was removed, and the membrane overlying the skull was cleared using a microblade (Roboz). An aluminum headpost was attached to the skull using RelyX luting cement (Henry Schein). The skin overlying the cheek and zygomatic bone was removed, and vessels over the zygomatic bone were sealed using a cauterizing iron (Fine Science Tools). The muscle above and attached to the zygomatic bone was peeled away, and the bone was removed with microscissors (Roboz). The membrane and muscle holding the jawbone and associated tissue in place were then slightly peeled back to allow access to the skull underneath. A dental drill was used to thin the bone directly overlying the lateral olfactory tract (LOT), and fine forceps (Fine Science Tools, USA) were used to remove the thinned skull and dura underneath. Using a micromanipulator and injection assembly kit (Narishige; WPI), 3000–3500 nL of rabies-GCaMP3 virus was slowly pressure injected via a pulled glass pipette at five locations; three approximately equidistant locations directly underneath the LOT (normal to the surface of the brain), and two locations ~500 μm deep to the surface of the brain in the anterior portion of the exposed area. The craniotomy was covered with silicone sealant (WPI), and the surgical exposure

was covered with a layer of lidocaine jelly (Henry Schein Veterinary) followed by a layer of silicone sealant. No signs of virus toxicity, such as highly fluorescent or blebbing cells could occasionally be observed before 9 days post-infection, but were clearly evident after 2 weeks post-infection. Therefore, mice all imaging experiments were performed 5–7 days post-infection.

## Histological processing

Animals were deeply anaesthetized with ketamine/xylazine and transcardially perfused with 10 ml PBS, followed by 10 ml 4% paraformaldehyde. The brain was removed and postfixed in 4% paraformaldehyde at 4°C overnight. A vibratome was used to cut 85 µm-thick coronal slices through the olfactory bulb, and slices were counterstained overnight in 1/1000 NeuroTrace 435 (Invitrogen) in PBS and mounted in Vectashield (Vector Labs) for imaging on a Zeiss 710 confocal microscope (Zeiss) using a 10x water immersion objective (Zeiss 0.45 NA).

## Olfactory bulb imaging

Mice were anaesthetized using ketamine/xylazine (100 mg/kg / 10 mg/kg, Sigma Aldrich) and the skull overlying the olfactory bulb was thinned using a dental drill and removed with forceps, and the dura was peeled back using fine forceps. A small circular coverslip cut from a cover glass (Corning #2870–18) using a diamond scriber (VWR) was placed over the exposed bulb and sealed in place using 2% agarose to minimize movement of the brain. Animals were then moved to a two-photon microscope (Ultima, Prairie Technologies, or Leica SP5) for imaging. A 16x objective at 2x zoom (Ultima) or a 25x (Leica SP5) was used to focus on the glomerular layer (~150 µm below the pial surface) or the mitral cell layer (~300–400 µm below the pial surface), and a Ti-Sapphire laser (Coherent) was tuned to 910 nm for experiments. Images (256 x 256 pixels) were acquired at a frame rate of 2.53 Hz (Ultima) or 2.9 Hz (Leica SP5). Odors were delivered at a flow rate of 1L/min for 2 s with inter-trial intervals of ~60 s. Odor stimuli for a given experiment consisted of one of two odor sets, delivered through a 16 channels olfactometer (Automate Scientific): a set of 13 monomolecular odorants (purchased from Sigma-Aldrich with the highest purity available) diluted at 1/10 000 vol./vol. dilution in mineral oil (Sigma-Aldrich), and a set of 'concentration series' consisting of three odors of ten fold increasing concentrations (1/100, 1/1 000 and 1/10 000 vol./vol. dilutions of acetophenone, ethyl acetate, and hexanone). Odorants were presented 4 times each, in pseudorandomized order. A total of 2–3 spatially distinct sites (often consisting of the posterior, medial, and anterior dorsal surface of the bulb) were imaged in each mouse.

## Electrophysiology

For anaesthetized recordings, male and female M71 transgenic mice and their littermate controls (6–9 weeks old) were anaesthetized intraperitoneally with ketamine and xylazine (100 mg/kg and 20 mg/kg, respectively for induction; xylazine concentration was reduced to 10 mg/kg for maintenance) and kept warm (37°C; DC temperature controller, FHC, Bowdoin ME, USA) for the duration of the experiments. A small craniotomy and duratomy of approximately 1–2 mm in diameter was made over the dorsal right olfactory bulb, which was submerged in Ringer solution containing (in mM): NaCl (135), KCl (5.4), HEPES (5), MgCl2 (1), CaCl2 (1.8), and its pH adjusted to 7.2 and 280 mOsm/kg. Whole-cell recordings were made with borosilicate glass pipette filled with (in mM): KMeSO4 (130), HEPES (10), KCl (7), $ATP_2$-Na (2), ATP-Mg (2), GTP (0.5), EGTA (0.05), biocytin (10), with pH and osmolarity adjusted to 7.3 and 275–80 mOsm/kg, respectively. Signals were amplified and filtered at 30 kHz by an Axoclamp 2B (Molecular Devices, Sunnyvale, CA, USA) and digitized at 20 kHz with a micro 1401 (Cambridge Electronic Design, Cambridge, UK). Odors were presented to the animals using a custom-made flow-dilution olfactometer at approximately 1% of saturated vapor with an inter-trial interval of 10 s (awake) or 20–25 s (anaesthetized). All recordings were done blindly with respect to the genotype of the animals. Data were analysed in Matlab (MathWorks, Natick, Massachusetts, USA). To calculate the evoked membrane potential (Vm) in anaesthetized animals, voltage traces were first aligned to expiration peaks of respiration rhythms (chest distension, see *Fukunaga et al., 2012*; *Schaefer et al., 2006*). The average waveform from the baseline period was subtracted from the aligned voltage trace from first complete sniff-cycle after odor valve opening. Evoked Vm for each cell was the mean of this subtracted component, averaged across trials. Responses were defined as significantly hyperpolarizing or depolarising if the evoked Vm deviated

by more than -2 or 2 standard deviations from baseline fluctuations, respectively. For two-sample KS test, the test statistic, D, was max $(| F_1(x) - F_2(x) |)$, where $F(x)$ is the cumulative distribution function for each dataset.

For awake recordings, head plate surgery and craniotomy was performed either directly preceding the electrophysiological recording or up to 2 days before (cf. *Kollo et al., 2014*) under isoflurane anaesthesia (5% for induction, 1.5–3% for maintenance in 95% oxygen), with local (0.5% mepivicaine s.c.) and general anaesthesia (5mg/kg carprofen, s.c.) administered. Recordings, solutions and analysis were as described above for anaesthetized animals. The only exception was that in awake animals, where sniff length is more variable, for each trial, the baseline Vm was calculated as the mean Vm during the 2 s prior to odor onset, and this was subtracted from the Vm during odor period. Evoked response was directly calculated as the mean Vm during the 1 s odor period, averaged over all trials. For FR responses, FR was calculated within each 0.25 s time bin aligned to the first inhalation post odor onset. Baseline FR was calculated on each trial as the mean within the 2 s prior to odor onset, and this was subtracted from the odor period. FR response was calculated as the mean FR in all time bins of the odor period across all trials. To test whether FR responses were significant, a paired T-test was performed between FR calculated during baseline in the 2 s prior to odor onset, and those calculated during the 1 s odor stimulus for all trials. Theta tuning was calculated from Vm during sniffs of durations >0.2 s and <0.32 s within the inter-trial intervals as described previously (*Fukunaga et al., 2012*).

## Imaging data analysis

Data analysis was conducted in ImageJ and Matlab. Motion artifacts were first corrected by using a subpixel translational-based discrete Fourier analysis (*Guizar-Sicairos and Fienup, 2008*). ROIs were then manually drawn on an average image of the imaging site, and the pixel gray values averaged in each ROI were used to estimate the fluorescence of single cells at each time frame. For each trial, the change in fluorescence ($\Delta F/F_0$)was calculated as $(F-F_0) / F_0$, where $F_0$ is the median value between seconds 2 and 6 of the pre-odor period. We estimated the baseline fluctuation at a given trial as the standard deviation (SD) of $\Delta F/F_0$ during the baseline period. Odor responses were assessed over a 10 s period following odor onset. A cell was deemed responsive if it exceeded response threshold (3.2 x SD for mitral cells, false positive rate (FPR) = 1.3%, 3.4 x SD for periglomerular cells, FPR = 3.8%) during at least 3 frames in this period. Using a more stringent response criterion for mitral cells (3.8 x SD) yielded reduced numbers of odor-responsive neurons, but did not change the relative distributions of odor-responsive neurons between M71 transgenic mice and controls, or mitral cell response variability (data not shown). The percent of responding neurons to each stimulus was calculated as the average number of active neurons across 4 trials. To construct the odor spot maps and to calculate the tuning curves, only cells that responded at least 2 out of the 4 trials were included. To build the cross-correlation matrix of the patterns of activity we combined $\Delta F/F_0$ responses of all mitral cells, averaged over the 4 s following odor onset into a single mitral cell x odor trial matrix. We then calculated the cross-trial correlations of the patterns of mitral cell activity.

## Statistics

All descriptive statistics in the text are mean ± SD. Before performing parametric statistical tests (ANOVA), homogeneity of variance within datasets was tested by computing the maximum variance ratio $Max(s^2)/Min(s^2)$ between groups. Homogeneity of variance was assumed if the maximum variance ratio was below 4. To explore the variability of mitral cell odor representation density across genotypes and odorants (*Figure 1H–I*), we used a mixed-effect ANOVA with genotype and odor as fixed-effect categorical factors, and imaging site as a random effect variable to account for repeated measure of the same imaging site in the course of an experiment. To quantify the differences in the concentration dependence of neural responses across stimuli and genotypes (*Figure 4*), we regressed the number of responsive neurons on stimulus intensity, and calculated the difference in linear change between stimuli ($\Delta LC = LC_{acetophenone} - LC_{ethyl\ acetate}$) as the difference between their regression slopes.

## Acknowledgements

We thank Ludovic Cacheux for help with imaging experiments and data analysis, Marion Ruinart de Brimont for genotyping of transgenic mice, Yves Dupraz for his work on the in vivo imaging set-up, Jérémie Teillon and Philippe Mailly for help with imaging data pre-processing, and France Maloumian for help in preparing the figures. We also thank Richard Axel for scientific discussions and support, and Gilad Barnea and Gabriel Lepoussez for critical comments on this manuscript. This work was supported by a Marie Curie International Reintegration grant (IRG 276869) and the "Amorçage de jeunes équipes" program (AJE201106) of the Fondation pour la Recherche Médicale (to AF), a postdoctoral fellowship by the LabEx "MemoLife" (to BR), and a NIH "Pathway to Independence" award, DC009839 (to KMF). Work in ATS's laboratory was supported by the Francis Crick Institute, which receives its core funding from Cancer Research UK, the Medical Research Council (MC_UP_1202/5) and the Wellcome Trust. This work was further supported by the Max Planck Society, DFG-SPP 1392 (to ATS), a Boehringer Ingelheim PhD fellowship (RJ), and the Alexander von Humbold Foundation (to IF). Initial imaging experiments were performed in Richard Axel's laboratory, which is supported by the Howard Hughes Medical Institute.

## Additional information

### Funding

| Funder | Grant reference number | Author |
|---|---|---|
| Fondation pour la Recherche Médicale | AJE201106 | Alexander Fleischmann |
| Medical Research Council | MC_UP_1202/5 | Andreas T Schaefer |
| Max-Planck-Gesellschaft | DFG-SPP 684 1392 | Andreas T Schaefer |
| Alexander von Humboldt-Stiftung | | Izumi Fukunaga |
| European Commission | IRG 276869 | Alexander Fleischmann |
| National Institutes of Health | DC009839 | Kevin M Franks |

The funders had no role in study design, data collection and interpretation, or the decision to submit the work for publication.

### Author contributions

BR, RJ, IF, Conception and design, Acquisition of data, Analysis and interpretation of data; DLS, AF, Conception and design, Acquisition of data, Analysis and interpretation of data, Drafting or revising the article; AD, Acquisition of data, Analysis and interpretation of data; IW, Contributed unpublished essential data or reagents; KMF, ATS, Conception and design, Analysis and interpretation of data, Drafting or revising the article

### Author ORCIDs

Alexander Fleischmann, http://orcid.org/0000-0001-7956-9096

### Ethics

Animal experimentation: All experiments were performed in accordance with approved institutional animal care and use committee protocols of Columbia University (#AC-AAAH9255), and in accordance with the INSERM Animal Care and Use Committee guidelines (#B750512/00615.02), the German Animal Welfare Act, and the UK Home Office and the Animals and Scientific Procedures Act (#PPL 70/7827).

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
