## [Decision Letter]

Thank you for submitting your article "Massive normalization of olfactory bulb output in mice with a "monoclonal nose"" for consideration by *eLife*. Your article has been favorably evaluated by a Senior editor and three reviewers, one of whom, Upinder S Bhalla, is a member of our Board of Reviewing Editors.

The following individual involved in review of your submission has agreed to reveal their identity: Naoshige Uchida.

The reviewers have discussed the reviews with one another and the Reviewing Editor has drafted this decision to help you prepare a revised submission.

The reviewers felt that there were some points about presentation of theresults that the authors should address.

1) The authors should explain the behavioral results more clearly, especiallyclarifying the design of the experiment and its interpretation.

2) Figure legends from in-vivo recordings should clearly state if therecordings are from awake or anesthetized animals.

3) The results regarding response variability should be reanalyzed withmore stringent odor response criteria to test if the main conclusionsremain the same. A more thorough discussion of response variability shouldalso be included.

By way of context, the full reviews are appended here. Please note that the essential revisions are in the 3 points of the synthesis above, but we felt that the fuller reviews may give some insights into the viewpoints of the reviewers.

*Reviewer 1:*

This is a substantially enhanced resubmission of an earlier manuscript that was considered at *eLife*. The authors have done several additional experimentsand analyses, strengthening the conclusions of the paper.

An initial experiment is to verify and extend earlier work on the inabilityof M71 transgenic mice to discriminate between acetophenone and mineral oildiluent. They confirm this, but extend the result by showing that the micedo detect acetophenone in an exploratory sniff behavior test. This is aninteresting additional nuance to the behavior, and shows that there is somekind of detectability still available to the system. This possibly sheds lighton the paradox of observed activity changes not appearing to map to behavior.

This would be an interesting line to follow up in future work.

The previously presented observations on the responses of mitral cellsin the wt and M71 cases remain strong, and are now extended in thisresubmission. There is better analysis, especially of concentration dependence.

There is an additional analysis of respiration tuning, which yields the interesting result of higher tuning in M71 mice than wild-type.

One of the key original concerns was the lack of in-vivo awake recordings,which the authors have now done. These support the original anesthetizedrecording results. The number of recorded neurons is somewhat small andthus limits the kinds of analysis possible, but the basic observations arewell supported by the new, awake recording data.

Overall I feel that this manuscript is now strong, and its conclusions well-supported.

*Reviewer 2:*

The authors have addressed my previous concerns adequately. I have no further concerns.

*Reviewer 3:*

The current version of the manuscript is significantly stronger. The authors followed many of the reviewer's suggestions. However, a few things still need some work.

1) Behavioral results.

The authors repeated behavioral experiments previously published in 2008 and got the same result. They also performed a new experiment, which was very revealing, and demonstrated that the monoclonal mice can detect acetophenone. However, the mystery still remains. In the Discussion, the authors tried to explain such a strange outcome. They proposed two potential explanations.

First, the presentation of acetophenone evokes a prolonged network perturbation, which may last "for several tens of seconds after stimulus offset", and such a perturbation interferes with the task. It was very difficult to make sense of this explanation, given that the description of the behavioral paradigm lacks details. For example, the authors stated that inter-trial interval was only 1-2 seconds? Why was it so short? Is it an obvious parameter to change given the proposed explanation? The behavioral parameters should be fully described in the methods section.

The second potential explanation is that an animal cannot discriminate a puff of acetophenone from a puff of clean air contaminated with acetophenone; i.e., even a very low concentration, acetophenone evoked sufficient MT cell activity, which was indistinguishable from activity at the higher concentration of the odorant. This is a very intriguing explanation with profound consequences. It means that monoclonal nose animals lack the ability to discriminate concentrations of the ligand, and their concentration dynamic range is very low. This result may inform us about concentration/intensity coding in the olfactory system. I wonder why the authors stopped short. Why in their imaging studies would they not lower acetophenone concentration to test this hypothesis? It would be ideal if they can present odors in their imaging studies using exactly the same odor delivery system used in the behavioral setup. The Knosys olfactometer may indeed suffer from a contamination problem, because it has C-flex tubing in the odor pathway.

The paper by itself is already standing at a high level and addresses multiple questions. It is significantly improved compared to the previous version. However, the authors are in a good position to make relatively small efforts and solve the mystery. This probably won't change the paper conclusion about massive renormalization in the olfactory bulb, and I would not insist on it. However, one of the main problems of systems level neuroscience is to relate the network phenomena to behavior, but the current behavioral experiments are not conclusive.

In any case, I strongly insist the authors make a much better description of the behavioral methods.

2) Awake vs. anesthetized.

Presenting the results from awake animals is a major improvement compared to the previous version. In the Methods section the authors described both e-phys recordings in awake and anesthetized states, but reading the manuscript it is difficult to sort out which experiments were performed in which state. Please make direct references in all figure captions if the data obtained in awake or anesthetized mice. In the Discussion, the authors should also address this question, specifically imaging data in the light of results from Kato 2012.

3) Response variability.

It is very intriguing result, but it is still hard to make sense of it.

First, the variability of control and monoclonal mice were measured in anesthetized state. It is not clear how these results can be extrapolated to functional significance. Kato 2012 results showed that MT cell responses in awake mice become sparser. Do they become more or less variable? Can the authors propose and discuss the sources of variability, and why it is different in control and experimental mice?

Second, the previous papers about Ca imaging in MT cells have different criteria for cell stimulus responses. Kato 2012 defined odor responses as "Odor-ROI pair was considered responsive only when its response exceeds threshold both 1) in at least 4 of the 7 trials and 2) in the trial-average trace." Livneh 2014 defines as "Normally we used for analysis all four repetitions, in rare cases we had to discard one trial." The way the authors defined the odor responses may lead to increase of variability estimates. The result is still valid: for a given response definition, the monoclonal mice have higher response variability than the control mice. However, if the authors choose stricter criteria for odor responses, the monoclonal mice, which have higher variability, may have lower number of odor responsive cells. This will change one of the central claims of the paper.

I may be missing something, but this subject definitely needs more analysis and discussion.

My overall conclusion is that the results are definitely worth publishing, but the paper needs more work.

[Editors’ note: a previous version of this study was rejected after peer review, but the authors submitted for reconsideration. The previous decision letter after peer review is shown below.]

Thank you for choosing to send your work entitled "Massive normalization of olfactory bulb output in mice with a "monoclonal nose"" for consideration at *eLife*. Your full submission has been evaluated by a Senior editor and three peer reviewers, one of whom is a member of our Board of Reviewing Editors, and the decision was reached after discussions between the reviewers. Based on our discussions and the individual reviews below, we regret to inform you that your work will not be considered further for publication in *eLife*.

The reviewers agreed that the monoclonal nose model of olfaction is very interesting and the questions on mechanisms for apparent similarity of M/T responses to odor between M71 mice and wildtype were important.

The three reviewers agreed that the conclusions of the paper were not adequately supported by the presented data. The key point of concern was that the conclusion of mitral cell response similarity between M71 and wild-type mice would probably change if the animal was awake. A further set of conclusions about the reliability of odor responses was also felt to be sensitive to the changed dynamics of responses in awake animals.

In addition to these primary concerns, the reviewers noted that the observations of a strong M/T response to acetophenone seems difficult to reconcile with the previously reported behavioral inability to detect M/T responses. They felt that confirmation of previous results on non-detection of acetophenone by M71 mice would be important to clarify this point, and would be of great interest to the field. Further behavioral support for the conclusion about reliability of odor responses would also be desirable. While the reviewers would like to see these results, these points were not felt to be crucial for the paper.

The reviewers also made a number of other suggestions for strengthening the paper.

*Reviewer #1:*

The study uses 'monoclonal nose' mice with GCAMP-3 imaging to look at how representations change. The key finding is that mitral cell responses are remarkably normal looking, except at high odor concentrations. This is despite the fact that 95% of sensory neurons express the M71 receptor. In some respects this kind of resilience is reminiscent of the Slotnick papers where odor discrimination remained intact despite lesions to 95% of the olfactory bulb.

1) The authors investigate the mechanisms for how the olfactory bulb circuitry achieves this level of robustness in function. They carry out patch recordings from mitral cells as well as calcium imaging of populations of mitral cells that project to the piriform. These recordings in wt animals are consistent with previous work, and the M71 recordings are very similar. The authors do identify a few interesting differences, in amplitude, duration and consistency of response. While the point about consistency of response is readily understood, as the authors say, if the circuit can "greatly amplify weak odor-evoked signals," it would be useful for the authors to suggest mechanisms for the other differences. It wasn't clear if the authors had used their patch recordings in the context of odor tuning. If so, this data might provide insights.

2) The analysis of the inhibition of acetophenone-evoked activity is interesting but possibly there are further insights to be gleaned from the available data. The authors build their case by combining intracellular mitral-cell recordings with Ca, imaging of PG cells, and then examine a concentration range for stimuli. I am intrigued that in Figure 4, the response of the mitral cells does not saturate by 1% even though that of PG cells saturates around 0.1%. The authors suggest in the Discussion that this may imply that there are further stages of negative feedback, possibly in the granule cell layer.

I wonder if there may be some clues to this proposed mechanism in the spatial distribution of the responsive mitral cells as the acetophenone concentration is raised. Another possible clue might come if the authors examine their data on trial-to-trial variability for their mitral cell responses. Do the authors have this data for a range of acetophenone concentrations?

3) Did the authors obtain patch recordings from mitral cells at a range of acetophenone concentrations? This too may provide insights about the origins and levels of negative feedback.

4) Unfortunately the authors do not estimate the respiration tuning in the imaging, as they use only a 2.5 Hz frame rate. However, the patch recordings from mitral cells could be used to examine respiration tuning. The authors have done some aligning to respiration in the analysis of evoked Vm. I feel that a systematic analysis of respiration tuning could be quite informative. For example, how consistent is the respiration tuning to acetophenone between different M/T cells in a wt transgenic? Is there a preferred phase of inhibition? This might be expected for bulk PG cell inhibition, but not if there were more complex circuit mechanisms operating.

*Reviewer #2:*

In this manuscript Sosulski et al., study a very interesting model of olfactory processing, a "monoclonal nose" mouse, which expresses a single olfactory receptor (OR) gene (M71) in 95% of olfactory sensory neurons (OSNs), while all other OR genes are expressed in the remaining 5% of OSNs. This work is a follow up to the first work describing this mouse (Fleischmann, 2008). The current paper attempts to answer a few interesting problems formulated in the first paper: 1) How, despite the drastic decrease in the number of OSNs expressing a given OR type (except M71), do these mice show no behavioral deficits in discriminating or detecting non-M71 ligands? 2) Why, despite overexpression of the M71 receptor, are these mice unable to detect an M71 ligand (acetophenone)? Both the imaging and electrophysiological data presented in this paper are promising; however, they do not answer either of these questions, and, I would say, do not add much to our knowledge about the olfactory bulb processing.

1) In the first paper (Fleischmann, 2008), it was proposed that feedforward inhibition could suppress the mitral/tufted (M/T) cells' responses in the absence of the contrast of glomeruli pattern, as an explanation for the deficit in behavioral response to M71 ligands. The current paper reports an obvious M/T cell response to acetophenone, and a strong concentration dependence to this ligand. How, then, do authors explain the inability of monoclonal mice to detect M71 ligands? The arguments about postsynaptic inhibition do not reconcile with their current findings, because the measured calcium activity in mitral cells is a putative proxy of spiking output. No matter the differences in bulbar circuitry and processing, if a somewhat normal signal is sent to cortex, the animal should be able to detect it.

While I will focus my comments on the current paper, some of its conclusions are based on data from the previous paper, published years ago. The result that an M71 monoclonal mouse cannot detect M71 ligands is very important, yet quite controversial. I would suggest that if the authors want to build on this observation, they should reproduce it. Given the current result that the mitral cells in M71 mice respond more or less normally to acetophenone, I harbor doubts that these mice would not be able to detect the odor. Either the previous behavioral result is incomplete or conclusions from anesthetized mice may not reflect the whole picture (see below).

2) One of the main arguments of the paper is that the statistics of mitral cell responses to non-M71 odors are similar in M71 and control mice. It has been shown that not only is the resting state of M/T cells quite different in awake and anesthetized states (Rinberg, 2006, but also see Kollo, 2014), but also that M/T cell odor responses become sparser in the awake state (Kato, 2012). Thus, the balance between excitation and inhibition in these two states is likely very different (see also Cazakoff, Nat.Neuro., 2014). Since it is unclear exactly how the balance differs in these M71 mice, it is hard to predict if M/T cell activity would transition in a similar way to control mice between anesthetized and awake states, or how well the anesthetized activity reflects the potential difference in network function that could emerge in the awake-state activity of these mice. Given that the current technology allows recording in awake mice, I do not see any reason why authors did not address the question more directly. Also, state-dependent changes in inhibition and M/T output were not even mentioned in the Discussion.

3) I found it very interesting that the 5% of OSNs carrying the odor signal can evoke a similar effect on postsynaptic cells, mitral cells, as a population. After reading this paper it is very hard to make definite conclusions about the mechanisms for such normalization. The authors proposed a role for PG cells in this process. It is one plausible explanation, but I am not sure how much this paper adds to revealing the mechanisms of signal normalization.

In my opinion, there are two basic approaches. First, that the proposed mouse model can help us understand basic mechanisms of olfactory processing. The data presented does not convince me that this is the case. Second, a new model by itself presents an interesting phenomenon, and it is worth understanding how, with such unnatural structure of receptor input, the system can behave in an almost "normal" way. I am not sure that this question has been answered here.

4) Another result is very intriguing: the reliability of the odor responses in M71 mouse is significantly lower than in a control mouse. The authors suggest that the decrease of the Pearson correlation coefficient from 0.4 to 0.17 tells us something about the decrease of the mouse's ability to discriminate odors. First, we do not know if the increased variability of the responses remains in awake mice. Second, the comparison to behavioral performance seems like a stretch. Do M71 mice perform poorer than control mice with the same odors for which variability was measured? The authors should either make a more direct connection between the physiological and behavioral observations, or they should explain the origin of such variability.

I think that the authors have a very interesting and promising system to work with. However, I would caution them to reproduce their behavioral result before building on it. Overall, the current observations are not conclusive enough to explain the phenomena of this altered system or to expand our knowledge about the olfactory system in general.

*Reviewer #3:*

In this manuscript, Sosulski and colleagues characterized the neural activity in the olfactory bulb (OB) of transgenic mice exogenously expressing the M71 odorant receptor (OR) gene in the vast majority of olfactory receptor neurons (ORNs). A previous study (Fleischmann et al., 2008) demonstrated that the expression of endogenous ORs was greatly reduced in M71 transgenic mice. Despite this alteration, these mice were able to detect and discriminate a variety of odors, with noticeable deficits in discriminating very similar odors. They also showed that these mice failed to detect acetophenone, an M71 receptor ligand, despite the fact that acetophenone activates the vast majority of ORNs and OB glomeruli. The neural basis of these behavioral deficits remains to be clarified.

In the present study, the authors characterized the neural activity in the OB in anesthetized animals using two-photon calcium imaging and whole-cell patch clamp recording. The authors demonstrate that the fractions of mitral cells activated by an odor were very similar between wild-type and M71 transgenic mice (~15% ) despite the great reduction in endogenous OR expressions in M71 transgenic mice. Interestingly, the trail-to-trial variability of odor responses was greater in M71 transgenic mice. The authors then measured the subthreshold membrane voltage dynamics with whole cell recording and showed that odor stimulation often evoked hyperpolarization in M71 transgenic mice, consistent with a role of inhibitory mechanisms in normalization. These results demonstrate that massive normalization of odor-evoked activities occurs at the level of the OB and the increased trial-to-trial variability may underlie the behavioral deficits. The latter idea is consistent with the idea that massive convergence of ORN axons increases the reliability of odor responses.

These findings are striking and provide important insights into the neural mechanisms underlying the behavioral phenotypes of M71 transgenic mice, which in turn reveals aspects of neural processing that occur in the OB. Overall, this work contains various findings that are of interest to those who are interested in the olfactory system.

---

## [Author Response]

Reviewer 3:

1) Behavioral results.

The authors repeated behavioral experiments previously published in 2008 and got the same result. They also performed a new experiment, which was very revealing, and demonstrated that the monoclonal mice can detect acetophenone. However, the mystery still remains. In the Discussion, the authors tried to explain such a strange outcome. They proposed two potential explanations.

First, the presentation of acetophenone evokes a prolonged network perturbation, which may last "for several tens of seconds after stimulus offset", and such a perturbation interferes with the task. It was very difficult to make sense of this explanation, given that the description of the behavioral paradigm lacks details. For example, the authors stated that inter-trial interval was only 1-2 seconds? Why was it so short? Is it an obvious parameter to change given the proposed explanation? The behavioral parameters should be fully described in the methods section.

[…]

*In any case, I strongly insist the authors make a much better description of the behavioral methods.* We agree with the reviewer that the go/no go conditioning tasks needs be to described in more detail, and we thank the reviewer for bringing this to our attention.

In our experiments, all mice were well trained to perform the task, i.e. they discriminated simple odor pairs (ethyl acetate, citronellol, carvone, in various combinations) with high accuracy. Individual experiments consisted of 200-300 trials, and experiments typically lasted for ~30 minutes, characterized by periods of intense odor sampling that were occasionally interrupted by periods of exploration of the conditioning chamber. Our previous comments about inter-trail intervals were indeed misleading. We have now detailed the structure of each behavioral trial in the Methods section. In brief, initiated by a light beam break by the animal, each trial starts with a 0.5 second preparation period to ensure rapid odor presentation, followed by a two second odor presentation. Mild suction ensures clearing of odors after each trial, but the system was not designed for rapid clearing. After a variable time interval chosen by the animal the next trial commences. In order to specifically compare the inter-trial interval with the post-odor exposure theta oscillations we have now analyzed inter-trial intervals for 7 animals. We find that the median time from the end of one odor presentation (closing of the odor valve) to the beginning of the next is 6.3 ± 0.5 seconds (25 percentile: 6 ± 0 sec, 75 percentile 8.7 ± 2.6 sec, mean and SD across 7 animals, 300 trials each). This is substantially lower than the prolonged theta oscillations we observe during our head-fixed in vivo whole cell recordings following acetophenone presentation, which last for at least 20 seconds. We did not attempt to change inter-trial intervals in this task, as enforced, longer inter-trial intervals can substantially alter odor discrimination learning (Bodyak and Slotnick, 1999; Walker and O’Connell, 1986). Controlled variation of the inter-trial interval simultaneous with physiological recordings probing subthreshold theta oscillations is indeed a possible next step in probing the hypothesis of how acetophenone perception might be perturbed, but would go substantially beyond the scope of this manuscript. To clarify these points, we have added a detailed description of the task in the Methods section of the revised manuscript.

2) Awake vs. anesthetized.

*Presenting the results from awake animals is a major improvement compared to the previous version. In the Methods section the authors described both e-phys recordings in awake and anesthetized states, but reading the manuscript it is difficult to sort out which experiments were performed in which state. Please make direct references in all figure captions if the data obtained in awake or anesthetized mice. In the Discussion, the authors should also address this question, specifically imaging data in the light of results from Kato 2012.*

We appreciate the reviewer’s suggestion to clarify this point. We now specify the experimental conditions (imaging in anesthetized mice, recordings in awake or anesthetized mice) at the beginning of each figure legend. We also reference Cazak_off_ et al., 2014 and Kato et al., 2012 to highlight the effects of anesthesia on olfactory bulb activity.

3) Response variability.

It is very intriguing result, but it is still hard to make sense of it.

First, the variability of control and monoclonal mice were measured in anesthetized state. It is not clear how these results can be extrapolated to functional significance. Kato 2012 results showed that MT cell responses in awake mice become sparser. Do they become more or less variable? Can the authors propose and discuss the sources of variability, and why it is different in control and experimental mice?

[…]

My overall conclusion is that the results are definitely worth publishing, but the paper needs more work.

We thank the reviewer for his/her suggestion to clarify this important point. We agree with the reviewer that defining appropriate criteria for mitral cell odor responses is critical to correctly interpret our calcium imaging results. We chose a response threshold of 3.2 x SD, based on a ROC analysis, yielding a true positive rate (TPR) of 92.4%. In response to the reviewer’s concern, we have reanalyzed our data with a more stringent response criterion (3.8 x SD of the baseline trace, TPR = 86%). We find that modifying the response threshold does not change the interpretation of our results: the percentage of odor-responsive mitral cells is indistinguishable between M71 transgenic mice and controls, but response variability is increased in M71 transgenic mice (Figure 7).

Author response image 1.Increasing threshold stringency does not change the comparison of the fraction of responsive mitral cells or their response variability between M71 transgenic mice and controls.(**A**) Mean fraction of neurons (horizontal line) responding to a given odorant at 0.01% vol./vol. dilution, with the response threshold set at 3.8 x SD, in control (black) and M71 transgenic (red) mice. Dots represent the fraction of responding cells for a given imaging site. Error bars = 95% CI of the mean. (**B**) Percent of neurons responding to 1, 2, 3, or 4 out of 4 odorant exposures in control (black) and M71 transgenic (red) mice. Note that the fraction of neurons responding on 4 out of 4 trials in strongly reduced in M71 transgenic mice. Error bars = SEM.**DOI:**
http://dx.doi.org/10.7554/eLife.16335.015

With the original response threshold set at 3.2 x SD, 12.2% ± 10.4 (SD across odorants and sites) of M71 transgenic mitral cells respond to odor, compared to 14.5% ± 11.2 in littermate controls (mixed-effect ANOVA (genotype x odorants), F_(12, 223)_ = 0.5, p = 0.91). With a response threshold set at 3.8 x SD, 9.9% ± 8.9%, of M71 transgenic mitral cells respond to odor, compared to 10.8% ± 11.2 in littermate controls (mixed-effect ANOVA (genotype x odorants), F_(12, 223)_ = 0.41, p = 0.956).

The variability of mitral cell odor responses remains high in M71 transgenic mice, compared to littermate controls. With the more stringent response threshold set at 3.8 x SD, the number of cells responding to 4 out of 4 presentations of a stimulus is 3.6% ± 3.7% for M71 transgenic mice, compared to 15.1 ± 12.6% in controls. Together, these results show that M71 transgenic and control mitral cell responses are similarly affected by a change in our thresholding criterion.

We chose not to use the averaged response across individual trials as a criterion to select cells in our analysis. While averaged responses are indeed commonly used to assess odor-evoked response significance in calcium imaging experiments, we believe that they may incorrectly bias the analysis by excluding variable, yet robustly responding neurons. Therefore, data representing mitral and periglomerular call activity reflect the percent of active neurons averaged across individual trials, or the raw, unthresholded ∆F/F values. The only exception are the spot maps of mitral cell responses in Figure 2 and Figure 6, where we chose to depict cells responsive in at least two out of 4 trials.

Finally, in accord with the reviewer’s suggestion, we now discuss the increased variability of mitral cell responses in M71 transgenic mice in the revised version of the manuscript. One important source of variability in neural responses to sensory stimuli is noise, and neural circuit mechanisms to reduce variability often rely on averaging neural signals carrying redundant information (Faisal et al., 2008). Sensory neurons expressing the same odorant receptor, and converging onto two glomeruli in the olfactory bulb provide a striking example of this principle. In M71 transgenic mice, the number of sensory neurons expressing a given odorant receptor are strongly reduced, thus limiting the power of averaging to reduce variability.

[Editors’ note: the author responses to the previous round of peer review follow.]

Reviewer #1:

1) The authors investigate the mechanisms for how the olfactory bulb circuitry achieves this level of robustness in function. They carry out patch recordings from mitral cells as well as calcium imaging of populations of mitral cells that project to the piriform. These recordings in wt animals are consistent with previous work, and the M71 recordings are very similar. The authors do identify a few interesting differences, in amplitude, duration and consistency of response. While the point about consistency of response is readily understood, as the authors say, if the circuit can "greatly amplify weak odor-evoked signals," it would be useful for the authors to suggest mechanisms for the other differences. It wasn't clear if the authors had used their patch recordings in the context of odor tuning. If so, this data might provide insights.

Our imaging experiments show that the amplitude of mitral cells responses to odorants is reduced in M71 transgenic mice, while the response duration is increased. While the reduced amplitude of response is readily explained by the substantially reduced input strength (Fleischmann et al., 2008), we can only speculate about the mechanisms underlying the increased response duration: One possible explanation for this observation could be that – as stimulus-driven glomerular input strength is reduced – this in turn might elicit levels of inhibition below those observed in wild-type mice, resulting in inefficient shutdown of – and thus prolonged – mitral cell responses. We now discuss these possibilities in the Discussion of the revised manuscript.

We have now performed whole cell recordings in mitral cells of awake mice and determined their responses to acetophenone, heptanal, ethyl acetate, hexanone, and an odorant mixture containing isoamyl acetate, 2-nonanone and cyclohexanol. Consistent with imaging data and electrophysiology in anaesthetized animals, our new experiments reveal that mitral cell responses in M71 transgenic mice are remarkably similar to littermate controls, including their odor-evoked firing rates. We do not observe any striking differences in odor tuning between M71 transgenic mice and littermate controls. However, the limited number of cell-odor pairs we can obtain in in vivo patch-clamp recordings precludes a systematic analysis of odor tuning.

2) The analysis of the inhibition of acetophenone-evoked activity is interesting but possibly there are further insights to be gleaned from the available data. The authors build their case by combining intracellular mitral-cell recordings with Ca, imaging of PG cells, and then examine a concentration range for stimuli. I am intrigued that in Figure 4, the response of the mitral cells does not saturate by 1% even though that of PG cells saturates around 0.1%. The authors suggest in the Discussion that this may imply that there are further stages of negative feedback, possibly in the granule cell layer.

*I wonder if there may be some clues to this proposed mechanism in the spatial distribution of the responsive mitral cells as the acetophenone concentration is raised. Another possible clue might come if the authors examine their data on trial-to-trial variability for their mitral cell responses. Do the authors have this data for a range of acetophenone concentrations?*

We have now examined the spatial distribution of mitral cell responses in M71 transgenic mice and controls (see Figure 8). We calculated the nearest neighbor distances for all cell-odor pairs that responded at least once to one of the stimuli (acetophenone, ethyl acetate, and hexanone, at three different concentrations). We used the average of this distance across responding neuron as a measure of clustering: the smaller the distance, the more clustered the responding neurons. We find that responses in M71 transgenic mice were slightly more clustered than in controls, consistent with the response maps presented in Figure 2. However, this difference did not reach statistical significance. An analysis of the trial-to-trial variability of mitral cell responses to acetophenone and ethyl acetate across the 100-fold concentration range was presented in the original Figure 4—figure supplement 1. However, we now recognize that the analysis was difficult to read and incomplete. We have therefore included an analysis of cross-trial correlations in the revised manuscript (Figure 3—figure supplement 1). We find that acetophenone-evoked patterns of mitral cell activity are highly variable at low concentration but become more consistent at higher concentrations. However, it is important to emphasize that the more reliable response patterns elicited by acetophenone at higher concentrations are comprised of substantially larger neural ensembles than those observed for other odorants.

To measure the spatial distribution of odor-responsive mitral cells we identified cells that responded at least once to each stimulus, and we calculated the distance to their nearest neighbor. The average of these distances yields an estimate for the clustering of responsive neurons. Consistent with the spotmaps shown in Figure 2, we observe a tendency of odor-responsive mitral cells of M71 transgenic mice to be more clustered than wild-type littermate controls. Dots = individual imaging sites. Error bars = SD.

Author response image 2.Clustering of mitral cell responses in controls and M71 transgenic mice.**DOI:**
http://dx.doi.org/10.7554/eLife.16335.016

*3) Did the authors obtain patch recordings from mitral cells at a range of acetophenone concentrations? This too may provide insights about the origins and levels of negative feedback.*

We have recorded mitral cell responses to acetophenone at 4 different concentrations. We find that acetophenone at high concentration elicits strong and prolonged inhibition in the majority of mitral cells. A more detailed statistical analysis of the concentration dependence of acetophenone responses is precluded by the limited number of cell-odor pairs we can obtain in in vivo patch-clamp recordings in awake mice and by the residual variability in odor-evoked responses. Importantly, however, we have previously shown, by using layer-specific optogenetic interference, that the slow, phasic odor- evoked inhibition we observe originates in the glomerular layer (Fukunaga et al., 2014).

*4) Unfortunately the authors do not estimate the respiration tuning in the imaging, as they use only a 2.5 Hz frame rate. However, the patch recordings from mitral cells could be used to examine respiration tuning. The authors have done some aligning to respiration in the analysis of evoked Vm. I feel that a systematic analysis of respiration tuning could be quite informative. For example, how consistent is the respiration tuning to acetophenone between different M/T cells in a wt transgenic? Is there a preferred phase of inhibition? This might be expected for bulk PG cell inhibition, but not if there were more complex circuit mechanisms operating.*

We thank the reviewer for this suggestion. We have now systematically analyzed respiration tuning in response to acetophenone in M71 transgenic mice. We find that theta coupling in response to acetophenone exposure is indeed massively amplified in M71 transgenic mice after acetophenone exposure. During baseline conditions, theta coupling of MTCs was slightly weaker in M71 transgenic mice. There was no significant difference in the preferred phase of M71 transgenic or control MTCs. During odor exposure there was no significant difference in phase preference. With respect to inhibitory responses it is difficult to ascertain preferred phase as inhibition tends to be strong, prolonged and seemingly unmodulated. We do now show the average acetophenone response for each cell recorded in the awake animal (Figure 5—figure supplement 1). One potential explanation would be that odor-evoked inhibition might clamp the membrane potential to the Cl^-^reversal potential at the site of the synaptic input, thus masking any modulation. At this point this is pure speculation and we thus refrain from further discussions in the revised manuscript.

Reviewer #2:

1) In the first paper (Fleischmann, 2008), it was proposed that feedforward inhibition could suppress the mitral/tufted (M/T) cells' responses in the absence of the contrast of glomeruli pattern, as an explanation for the deficit in behavioral response to M71 ligands. The current paper reports an obvious M/T cell response to acetophenone, and a strong concentration dependence to this ligand. How, then, do authors explain the inability of monoclonal mice to detect M71 ligands? The arguments about postsynaptic inhibition do not reconcile with their current findings, because the measured calcium activity in mitral cells is a putative proxy of spiking output. No matter the differences in bulbar circuitry and processing, if a somewhat normal signal is sent to cortex, the animal should be able to detect it.

While I will focus my comments on the current paper, some of its conclusions are based on data from the previous paper, published years ago. The result that an M71 monoclonal mouse cannot detect M71 ligands is very important, yet quite controversial. I would suggest that if the authors want to build on this observation, they should reproduce it. Given the current result that the mitral cells in M71 mice respond more or less normally to acetophenone, I harbor doubts that these mice would not be able to detect the odor. Either the previous behavioral result is incomplete or conclusions from anesthetized mice may not reflect the whole picture (see below).

Following the reviewer's suggestion, we have repeated our previous experiments and now provide an additional set of behavioral data that confirm our initial observations (Figure 1, and Figure 1—figure supplement 2). We have also added a more quantitative description of these findings, which was not provided in the original publication, and show the raw data for each animal in the supplementary figure. These data recapitulate our original findings that M71 transgenic mice, which otherwise perform well on a go/no go operant conditioning odor discrimination task, do not discriminate between acetophenone and air in the same task.

However, we take the reviewer’s comment that failure to discriminate odor from air in such as task is not equivalent to stating that the animal cannot detect the ligand. We have also performed a new and different type of behavioral experiment that assesses the ability of M71 transgenic mice to detect odors. In the somewhat cruder, spontaneous task (sniff modulation in response to novel odor exposure), we find that M71 transgenic animals respond to both general odors and to acetophenone. This finding indicates that the previous interpretation that M71 transgenic mice are generally and completely unable to detect acetophenone was indeed an over-simplification.

Again following the reviewer’s suggestion, we performed whole cell recordings in awake mice. The finding that phasic odor responses to be acetophenone and other odorants are similar in wild-type and M71 transgenic mice are consistent in both awake and anesthetized mice. However, these new experiments also revealed a striking amplification of acetophenone-evoked theta oscillations. We hypothesize that these long-lasting (tens of seconds) and massively increased oscillatory network perturbations may impair the discrimination of acetophenone from air in the go/no go operant conditioning task. Therefore, these findings, together with the new result that sniffing is altered in response to acetophenone, require a more sophisticated explanation for our original findings that M71 transgenic mice do not discriminate between acetophenone and air in the go/no go operant conditioning task. We are grateful to the reviewer for this demanding and ultimately important critique.

2) One of the main arguments of the paper is that the statistics of mitral cell responses to non-M71 odors are similar in M71 and control mice. It has been shown that not only is the resting state of M/T cells quite different in awake and anesthetized states (Rinberg, 2006, but also see Kollo, 2014), but also that M/T cell odor responses become sparser in the awake state (Kato, 2012). Thus, the balance between excitation and inhibition in these two states is likely very different (see also Cazakoff, Nat.Neuro.,2014). Since it is unclear exactly how the balance differs in these M71 mice, it is hard to predict if M/T cell activity would transition in a similar way to control mice between anesthetized and awake states, or how well the anesthetized activity reflects the potential difference in network function that could emerge in the awake-state activity of these mice.

We agree with reviewer 2 that the balance of excitation and inhibition in the olfactory bulb is crucial to understanding the normalization of sensory input strength we observe in M71 transgenic mice, and that it is often believed that this balance is strongly affected by anesthesia. However, we disagree with the notion that olfactory bulb activity is vastly different between anaesthetized and awake, and we have indeed demonstrated that this is not the case when the entire population of cells is taken into account in an unbiased way, and if activity is analyzed on behaviorally relevant time scales of <2 seconds (Kollo et al, 2014). However, we recognize that a more detailed characterization of mitral cell responses in awake M71 transgenic mice can provide important new mechanistic insight intoolfactory bulb signal normalization. Therefore, to directly test for potential differences between awake and anaesthetized recordings we have performed whole cell recordings in awake mice to complement our imaging data in anesthetized mice. In addition, this new set of experiments allows for a detailed analysis of the temporal response properties of mitral cells while avoiding the biases associated with extracellular recording. The main findings that emerge from these new experiments are:

A) The basic physiological properties of mitral cells, including their input resistance, time constant and spontaneous firing rates are similar between M71 transgenic mice and controls.

B) In M71 transgenic mice, inhibition in response to acetophenone is increased and prolonged. In contrast, inhibition in response to non-acetophenone odorants is decreased.

C) Theta modulation in response to acetophenone is massively increased in M71 transgenic mice.

D) Mitral cell firing rates in M71 transgenic mice in response to acetophenone and non-acetophenone odorants are similar to wild-type.

E) Most parameters of mitral cell activity, including spontaneous and odor-evoked firing rates are similar in awake and anesthetized mice. Consistent with our published data (Kollo et al., 2014), anesthesia primarily appears to affect mitral cell response durations. No differences in the effects of anesthesia were observed between M71 transgenic and wild-type littermate controls.

*3) I found it very interesting that the 5% of OSNs carrying the odor signal can evoke a similar effect on postsynaptic cells, mitral cells, as a population. After reading this paper it is very hard to make definite conclusions about the mechanisms for such normalization. The authors proposed a role for PG cells in this process. It is one plausible explanation, but I am not sure how much this paper adds to revealing the mechanisms of signal normalization. In my opinion, there are two basic approaches. First, that the proposed mouse model can help us understand basic mechanisms of olfactory processing. The data presented does not convince me that this is the case. Second, a new model by itself presents an interesting phenomenon, and it is worth understanding how, with such unnatural structure of receptor input, the system can behave in an almost "normal" way. I am not sure that this question has been answered here.*

We agree with reviewer 2 that it is important to understand how olfactory bulb circuits can extract meaningful information from massively degraded sensory input. We believe that our manuscript, including new whole cell recordings in awake mice, substantially extends our mechanistic understanding of signal normalization in the olfactory bulb. Therefore, particularly in light of the new behavioral data and the extended electrophysiological data, we provide a considerably more refined mechanistic explanation for the behavioral phenotype of the model system.

*4) Another result is very intriguing: the reliability of the odor responses in M71 mouse is significantly lower than in a control mouse. The authors suggest that the decrease of the Pearson correlation coefficient from 0.4 to 0.17 tells us something about the decrease of the mouse's ability to discriminate odors. First, we do not know if the increased variability of the responses remains in awake mice. Second, the comparison to behavioral performance seems like a stretch. Do M71 mice perform poorer than control mice with the same odors for which variability was measured? The authors should either make a more direct connection between the physiological and behavioral observations, or they should explain the origin of such variability.*

The reviewer is correct that we cannot claim that the observed mitral cell trial-to-trial variability directly causes the odor mixture discrimination deficits observed in M71 transgenic mice. Rather, we state that these data are consistent with poorer performance. Although Reviewer 1 found this connection *‘readily understood’* and Reviewer 3 agrees that ‘[this variability] is consistent with the idea that massive convergence of ORN axons increases the reliability of odor responses’we have now clarified the link between this observation and the behavior in the text. Unfortunately, the limited numbers of cell-odor pairs that can be obtained in in vivo patch-clamp recordings in awake mice precludes a reliable statistical analysis of trail-to-trial variability under these conditions.